# SELECTIVE CLASSIFIER ENSEMBLE

## ABSTRACT

Selective classification allows a machine learning model to abstain from predicting some hard inputs and thus improve the safety of its predictions. In this paper, we study the ensemble of selective classifiers, i.e. *selective classifier ensemble*, which combines several weak selective classifiers to obtain a more powerful model. We prove that under some assumptions, the ensemble has a lower *selective risk* than the individual model under a range of coverage. The proof is nontrivial since the selective risk is a non-convex function of the model prediction. The assumptions and the theoretical result are supported by systematic experiments on both computer vision and natural language processing tasks. A surprising empirical result is that a simple selective classifier ensemble, namely, the *ensemble model with maximum probability as confidence*, is the state-of-the-art selective classifier. For instance, on CIFAR-10, using the same VGG-16 backbone model, this ensemble reduces the AURC (Area Under Risk-Coverage Curve) by about 24%, relative to the previous state-of-the-art method.

## 1 INTRODUCTION

Although recent years have witnessed the broad applications of deep learning models, their securities have not been fully guaranteed, which gives rise to the study of selective classification. For any given deep learning classifier, there might be inputs that the model is not able to classify in practical applications, for which the model might make unpredictable errors. To prevent this kind of error, we must accurately delimit the deep learning classifier's application scope. This need gives rise to the study of *selective classification* that learns a *selective classifier* $(f, g)$, where $f$ is a conventional classifier, and $g$ is a *selective function* that decides whether the selective classifier should abstain from prediction. Since the classifier is well studied, the study of selective classification focuses on the design of the selective function.

A standard approach to designing the selective function is to design a confidence score function with a threshold, and several confidence score functions have been developed. A simple confidence score function is the maximum predictive probability of the classifier (Hendrycks & Gimpel, 2017). More advanced methods modify the model architecture (Geifman & El-Yaniv, 2019) or the loss function (Liu et al., 2019; Huang et al., 2020) of the classifier to train the confidence score function and the classifier simultaneously. For example, Deep Gambler (Liu et al., 2019) regards the selective classification problem as gambling and proposes a novel loss function to train the classifier and the confidence score function. Although there are various individual models for the selective classifier, there has been no systematic study of the ensemble method in selective classification.

It is well known that the ensemble method, which combines the individual models to obtain a more powerful model, can improve the predictive performances of machine learning models (see Zhou (2012) for a review), but only a particular selective classifier ensemble, the ensemble of Softmax Response (Hendrycks & Gimpel, 2017), has been empirically studied by Lakshminarayanan et al. (2017). Ensembles of other kinds of selective classifiers, and the theoretical foundation of the ensemble in selective classification have not been studied yet. In this paper, we first demonstrate the theoretical foundation of the ensemble on selective classifiers, that is, with some assumptions, the ensemble has a lower selective risk than the individual model under a range of coverage. The proof is nontrivial since the selective risk (with the 0/1 loss) are non-convex. Second, we show the experimental results of the ensemble's performance in selective classification. The contributions of this paper are summarized as follows.

- We are the first to theoretically demonstrate that based on several reasonable assumptions, the ensemble has a lower selective risk than the individual model under a range of coverage. We verify this by systematic experiments on the tasks of image classification and text classification.

- We show a surprising experimental result that two simple methods, the SR ensemble and the Reg-curr ensemble, which can be summarized as *the ensemble model with maximum probability as confidence*, are the state-of-the-art selective classifiers.

## 2 PROBLEM FORMULATION OF SELECTIVE CLASSIFICATION

A selective classifier is composed of a standard classifier and a selective function. Considering a standard classification problem, $\mathbb{X}$ is a feature space, $\mathbb{Y} = \{1, 2, ..., K\}$ is a finite label set, and a classifier $f$ is a function $f : \mathbb{X} \to \mathbb{Y}$. A labeled dataset $\mathbb{D} = \{(x_i, y_i)\}_{i=0}^{N} \subseteq \mathbb{X} \times \mathbb{Y}$ is sampled from a distribution $p_{X,Y}$. Our goal is to learn a *selective classifier* where $f$ is a standard classifier and $g : \mathbb{X} \to \{0, 1\}$ is a selective function that estimates the correctness of $f$'s prediction. Given input $x$, the output of selective classifier $(f, g)$ is

$$(f, g)(x) = \begin{cases} f(x), & \text{if } g(x) = 1 \\ \text{Abstain}, & \text{if } g(x) = 0 \end{cases}. \tag{1}$$

Usually, $g$ is realized by a *confidence score* $\hat{\kappa} : \mathbb{X} \to \mathbb{R}^+$ with a threshold $\tau$ (Geifman & El-Yaniv, 2017), namely

$$g(x) = \mathbb{I}\{\hat{\kappa}(x) > \tau\}, \tag{2}$$

where $\mathbb{I}$ is the indicator function.

*Coverage* and *selective risk* are two basic evaluation metrics of selective classifiers, and the goal of selective classifiers is to minimize the selective risk for target coverage. The *coverage* of $(f, g)$ is defined to be the probability of $(f, g)$ not abstaining from prediction (Geifman & El-Yaniv, 2017), i.e.

$$\phi(f, g) := \mathbb{E}_{p(x)}[g(x)], \tag{3}$$

where $p(x)$ is the probability density function of input $x$. The *selective risk* (Geifman & El-Yaniv, 2017) of $(f, g)$ is

$$R(f, g) := \frac{\mathbb{E}_{p(x)}[\ell(f(x), y)g(x)]}{\mathbb{E}_{p(x)}(g(x))}, \tag{4}$$

where $\ell : \mathbb{Y} \times \mathbb{Y} \to \mathbb{R}^+$ is a given loss function. Usually, $\ell$ is the 0/1 loss (Geifman & El-Yaniv, 2017; 2019; Liu et al., 2019; Huang et al., 2020). Based on these definitions, the objective of selective classifiers is formalized as

$$\min R(f, g), \text{ s.t. } \phi(f, g) \geq c_{\text{target}},$$

where $c_{\text{target}}$ is a given target coverage.

When the selective function $g$ is developed as (2), the confidence threshold $\tau$ controls the trade-off between coverage and selective risk. With different values of $\tau$, $(f, g)$ has different pairs of coverage and selective risk $(\phi(f, g; \tau), R(f, g; \tau))$, which forms the *risk-coverage curve* (Geifman & El-Yaniv, 2017) of $(f, g)$. The risk-coverage curve specifies the entire performance profile of a selective classifier, and it is easy to see that the selective classifier with a lower risk-coverage curve is better. To evaluate selective classifiers more concisely, the *area under the risk-coverage curve* (*AURC*) is introduced as a metric of selective classifiers (Xin et al., 2021), and the selective classifier with a lower AURC is better.

## 3 RELATED WORK

Here, we summarize the previous studies on selective classification and ensemble methods. We also discuss the difference between selective classification and out-of-distribution detection.

## 3.1 SELECTIVE CLASSIFICATION

The critical problem of designing a selective classifier is to design its selective function, and there are two types of selective function $g$. One is the *implicit selective function*, which is derived from the classifier. The other is the *explicit selective function*, a neural network trained with the classifier $f$ simultaneously. Previous works for selective classifiers are listed as follows.

Selective classifiers with implicit selective functions include SR (Softmax Response) (Hendrycks & Gimpel, 2017), MC-Dropout (Monte Carlo-Dropout) (Gal & Ghahramani, 2016), and Reg-curr (Xin et al., 2021). In SR, the selective classifier is a vanilla classifier with its maximum predictive probability as confidence (i.e., the maximum output of the softmax layer). In MC-Dropout, it enables the dropout layer of the classifier and runs multiple feed-forward iterations at inference time to obtain the variance of the maximum probability output of $f$'s softmax layer, whose negative value is used as the confidence score. Reg-curr behaves the same as the SR at inference time but uses an RPP-based regularizer at training time, where RPP (Reversed Pair Proportion) (Xin et al., 2021) is the proportion of reversed pairs of confidence scores.

Selective classifiers with explicit selective functions include SN (SelectiveNet) (Geifman & El-Yaniv, 2019), Gambler (Deep Gambler) (Liu et al., 2019), and SAT (Self-Adaptive Training) (Huang et al., 2020). SN is a neural network that combines $f$ and $g$, where $f$ and $g$ share convolutional layers and have their separate fully-connected layers. The loss function of the model is the selective risk with some regularizers. A hyperparameter $c$ is needed to specify the target coverage. Gambler adds the abstention option to the classifier as an extra class, that is, for a given input $x$, the predictive probability of the extra class is the confidence of abstention, i.e., $1 - \hat{\kappa}(x)$. At training time, it regards the selective classification problem as gambling and is trained to maximize the gambling reward. Similar to Gambler, SAT adds the abstention option to the classifier as an extra class. However, SAT has a different training procedure, which is trained with a soft label that tells the model which sample to reject. SAT is the previous state-of-the-art selective classifier for image classification tasks.

## 3.2 RANDOMIZATION-BASED ENSEMBLE

In the *randomization-based* ensemble method, each member model is trained independently with the same architecture and training procedure but with different randomization seeds for the random initialization of parameters and the random shuffling of training data for each training epoch. At inference time, the predictive probability of the ensemble for each class is the average of those of member models. Lakshminarayanan et al. (2017) applies this ensemble method to deep neural networks (Deep Ensemble) and has achieved state-of-the-art performance in uncertainty estimation. The difference between Lakshminarayanan et al. (2017) and our work is that they only empirically study the ensemble of vanilla classifiers (the SR ensemble), but we not only empirically study ensembles of multiple outstanding selective classifiers but also provide the theoretical foundation of the ensemble in selective classification. As for the theoretical works, Krogh & Vedelsby (1994) proposes the *error-ambiguity decomposition* to explain the better performance of the randomization-based ensemble in regression tasks. However, for classification tasks, there is no such simple and elegant analysis, since the evaluation metrics are non-convex (Zhou, 2021). Thus, the corresponding analysis for classification tasks needs additional assumptions, e.g., unbiased, uncorrelated, and identically distributed estimation errors for the posterior probability distribution (Tumer & Ghosh, 1996; Fumera & Roli, 2005). Nevertheless, these assumptions are impractical (Fumera & Roli, 2005). As far as we know, there is no systematic study of the ensemble in the context of selective classification.

## 3.3 OUT-OF-DISTRIBUTION DETECTION

A related topic of selective classification is out-of-distribution (OOD) detection (Lakshminarayanan et al., 2017)) (also called as open set recognition (Scheirer et al., 2012), or novelty detection (Schölkopf et al., 2001)), which detects samples that differ significantly from a given dataset, i.e., OOD samples. The essential difference between selective classification and OOD detection lies in their different goals. The goal of the former is to detect samples where the classifier predicts incorrectly, which depends on both the classifier and samples, while that of the latter is to detect samples that differ significantly from a given dataset, which depends on samples only. In addition, at present, selective classification assumes that test data and training data are sampled from the same distribution (Geifman & El-Yaniv, 2017), instead of using OOD test data as OOD detection. Thus,

selective classification and OOD detection are complementary in preventing error predictions of machine learning models, as Figure 3 shows.

## 4 METHOD

With the randomization-based ensemble method, we propose the *selective classifier ensemble*. The basic idea is that each predictive probability (as well as the confidence score in the case of explicit selective functions) of the ensemble should be the average of those of the member models. Formally, we assume that for an input sample $x$, a classifier $f$ at first provides the predictive probability distribution $\hat{\boldsymbol{\pi}}_\theta = (\hat{\pi}_\theta^1, \cdots, \hat{\pi}_\theta^K)$ and then makes prediction $f(x; \theta) = \arg\max_{1 \leq k \leq K} \hat{\pi}_\theta^k(x)$, where $K$ is the number of classes, $\theta$ denotes the parameters of $f$, and $\hat{\pi}_\theta^k(x)$ is the predictive probability for class $k$ (the superscript is not an exponent). Then, the predictive probability distribution of the ensemble classifier of $M$ member models is

$$\hat{\boldsymbol{\pi}}_{\text{ens}}(x) := \frac{1}{M} \sum_{m=1}^{M} \hat{\boldsymbol{\pi}}_m(x). \tag{5}$$

The ensemble of the selective function is defined as follows. For implicit selective functions (e.g. SR), to keep the ensemble the same kind of selective classifier as the individual model (for example, the ensemble of SR should still be an SR model), the confidence score of the ensemble is derived from $\hat{\boldsymbol{\pi}}_{ens}$ in the same way as the individual model. For example, the confidence score of the *SR ensemble* is

$$\hat{\kappa}_{\text{ens}}(x) = \max_k \hat{\pi}_{\text{ens}}^k(x). \tag{6}$$

For explicit selective functions (e.g. SAT), the confidence score of the ensemble is the average of those of member models,

$$\hat{\kappa}_{\text{ens}}(x) = \frac{1}{M} \sum_{m=1}^{M} \hat{\kappa}_m(x). \tag{7}$$

## 5 THEORETICAL ANALYSIS OF SELECTIVE CLASSIFIER ENSEMBLE

In this section, we analyze the selective risk (with the 0/1 loss) of the ensemble of a simple selective classifier, the SR ensemble (see Section 4 for its definition). If the selective risk is a convex function of the predictive probability distribution, then according to the definition of the convex function, (6) implies that the selective risk of the ensemble is less than or equal to that of the individual model. However, the selective risk is non-convex because the 0/1 loss is a step function. Thus, the analysis is not easy. We need some assumptions to prove a lower selective risk of the ensemble. We introduce the assumptions in Section 5.1 (verified in Section 6.1) and the theoretical results in Section 5.2. The analysis for the other selective classifiers is left for future study.

### 5.1 ASSUMPTIONS

Given an SR ensemble with $M$ ($M > 1$) members, we assume that there are samples on which all member models provide almost the same predictive probability distributions. Furthermore, we idealize them as *definite samples*, for which all member models provide precisely the same predictive probability distribution. Then, the rest samples are referred to as *ambiguous samples*. Let $D$ be the event that the input sample is definite and $A$ be the event that the input sample is ambiguous. Considering that the input sample is randomly drawn from a dataset, the predictive probability for class $k$ ($1 \leq k \leq K$) and the confidence of the SR model are random variables. We denote these random variables as $\Pi^k$ and C respectively and use $\pi^k$ and $\kappa$ to denote their values respectively. Generally, for any continuous random variable Z, $p_Z$ denotes the *probability density function* (*PDF*) of Z, and for any real variable $z$, $z \to 1^-$ denotes that $z$ approaches 1 from the left. Based on the idealization and notations above, we introduce the following assumptions.

**Assumption 1.** For any individual SR model, with its confidence score denoted as C, we have

$$\lim_{\tau \to 1^-} \Pr(Err|A, \mathrm{C} \geq \tau) > \lim_{\tau \to 1^-} \Pr(Err|D, \mathrm{C} \geq \tau), \tag{8}$$

where $Err$ is the event that the model makes an error prediction.

In Assumption 1, $\Pr(Err|A, \mathrm{C} \geq \tau)$ is the selective risk of the individual model with a confidence threshold of $\tau$ for ambiguous samples, and $\Pr(Err|D, \mathrm{C} \geq \tau)$ is that for definite samples. The motivation for Assumption 1 is that ambiguous samples seem more difficult to classify than definite samples.

**Assumption 2.** For any individual SR model, both $\lim_{\kappa \to 1^-} p_{\mathrm{C}}(\kappa|D)$ and $\lim_{\kappa \to 1^-} p_{\mathrm{C}}(\kappa|A)$ exist and are non-zero, i.e., positive, where C is the confidence score of the SR model.

Assumption 2 claims that for both ambiguous samples and definite samples, when the confidence score of the individual model approaches 1, its PDF approaches a non-zero value. In other words, the individual model is *not modest* over both ambiguous samples and definite samples.

**Assumption 3.** For any SR ensemble of $M$ member models, let $\Pi_i^k$ be the predictive probability of the $i$-th member model for class $k$. Then $\forall k \in \{1, \ldots, K\}$, $p_{\Pi_1^k, \Pi_2^k, \ldots, \Pi_M^k}(\cdot|A)$, the joint probability density function of $\Pi_1^k, \Pi_2^k, \ldots, \Pi_M^k$ given the input sample being ambiguous, is bounded.

Assumption 3 is related to the definition of ambiguous samples and can be understood via the following example. Consider an ensemble with two members $\theta_1$ and $\theta_2$. Let their predictive probability for class $k$ be $\Pi_1^k$ and $\Pi_2^k$, $k \in \{1, 2, ..., K\}$, respectively. On definite samples, both $\theta_1$ and $\theta_2$ provide the same predictive probability distribution ($\Pi_1^k = \Pi_2^k, \forall k \in \{1, 2, ..., K\}$). Thus, for all $k$, $p_{\Pi_1^k, \Pi_2^k}(\cdot|D)$, the joint distribution of $\Pi_1^k$ and $\Pi_2^k$ given the input samples being definite, collapses to $\{(\lambda, \lambda)|\lambda \in [0, 1]\}$. In other words,

$$p_{\Pi_1^k, \Pi_2^k}(u, v|D) = \begin{cases} +\infty, & \text{if } (u, v) \in \{(\lambda, \lambda)|\lambda \in [0, 1]\} \\ 0, & \text{otherwise} \end{cases}.$$

On the contrary, ambiguous samples do not have such a property. We intensify this by Assumption 3 to provide a good analytical property of ambiguous samples. Furthermore, Assumption 3 reflects the diversity of the ensemble over ambiguous samples. Still consider the example above. If the predictions of $\theta_1$ and $\theta_2$ are sure to coincide, i.e., the ensemble model has no diversity, then the PDF of $\Pi_1^k$ and $\Pi_2^k$ is unbounded. Conversely, if the PDF of $\Pi_1^k$ and $\Pi_2^k$ is bounded, then the predictions of the member models are diverse. Thus, Assumption 3 provides the diversity of the ensemble over ambiguous samples. It is well known that the randomization-based ensemble has diversity (Zhou, 2012). Since the ensemble does not have diversity over definite samples, it must have diversity over ambiguous samples. Thus, we do not provide experimental verification for Assumption 3.

## 5.2 Analysis Results

With the assumptions above, we derive Theorem 2 (see Appendix B for proof details), which shows that the selective risk of the ensemble is lower than that of the individual model under a range of coverage. The intuition of its proof is as follows. According to Assumption 1, the individual model is **not** modest over ambiguous samples. On the contrary, based on Assumption 3, we prove that the ensemble is modest over ambiguous samples (Proposition 1). In addition, both the individual model and the ensemble are not modest over definite samples (due to Assumption 2 and the definition of definite samples). Thus, considering that the classifier's error rate over ambiguous samples is higher than that over definite samples when confidence approaches 1 (Assumption 1), the individual model suffers more wrong but confident predictions that come with ambiguous samples than the ensemble. Therefore, the selective risk of the individual model is higher than that of the ensemble (Theorem 2). In a word, the intuition is that because the ensemble avoids to be overconfident over ambiguous samples, the ensemble has a lower selective risk.

Before Theorem 2, we discuss Proposition 1, which provides critical insight into the better performance of the ensemble.

**Proposition 1.** *If Assumption 3 holds, then* $\lim_{\kappa_{\mathrm{ens}} \to 1^-} p_{\mathrm{C}_{\mathrm{ens}}}(\kappa_{\mathrm{ens}}|A) = 0$, *where* $\mathrm{C}_{\mathrm{ens}}$ *is the confidence score of the ensemble.*

Proposition 1 suggests that given the input sample being ambiguous, when the confidence of the ensemble approaches one, its PDF approaches zero. In other words, the ensemble is *modest* over ambiguous samples. Based on this proposition, we prove that the SR ensemble has a lower selective risk than an SR individual model under a range of coverage.

**Theorem 2.** *If Assumption 1-3 holds, then for any individual SR model and any SR ensemble, $\exists \phi_0 \in (0, 1)$ such that $\forall \phi \in (0, \phi_0)$,*

$$R_{\mathrm{ens}}(\phi) < R_{\mathrm{ind}}(\phi), \tag{9}$$

*where $R_{\mathrm{ens}}(\phi)$ and $R_{\mathrm{ind}}(\phi)$ are the selective risks of the SR ensemble and the individual SR model under coverage $\phi$, respectively[1].*

## 6 EXPERIMENTS

**Datasets.** We conduct experiments on multiple datasets for image classification and text classification tasks. Following Geifman & El-Yaniv (2017; 2019); Liu et al. (2019); Huang et al. (2020), we use CIFAR-10, CIFAR-100 (Krizhevsky, 2009), and SVHN (Netzer et al., 2011) for image classification tasks, and following Xin et al. (2021), we use MRPC (Dolan & Brockett, 2005), MNLI (Williams et al., 2018), and QNLI (Wang et al., 2018) for text classification tasks. The usage of the training set and test set in selective classification are the same as the standard classification, because current selective classification focuses on in-domain data (i.e., data from the same distribution as the training set) (Geifman & El-Yaniv, 2017). For example, if the selective classifier is trained on the training set of CIFAR-10 (Krizhevsky, 2009), then it will be tested on the test set of CIFAR-10. Furthermore, MNLI's development set and test set are divided into *matched* and *mismatched* parts. The matched parts are sampled from the same source as the training set (so they are in-domain data), while the mismatched parts are sampled from different sources. In our experiments, only the matched parts are used since the current selective classification only considers in-domain data. In addition, test sets of MRPC, QNLI, and MNLI are not accessible, so we use their development sets as test sets. Following Liu et al. (2019); Huang et al. (2020), since CIFAR-10, CIFAR-100 and SVHN originally had no development set, their development sets were 2000 samples randomly split from corresponding test sets. More details of all the datasets in our experiments are described in Appendix C.1.

**Evaluation Metrics.** The evaluation metrics are AURC and selective risk (the lower, the better for both). AURC is a comprehensive metric of selective classifiers, and selective risk is a standard metric in previous works (Geifman & El-Yaniv, 2019; Liu et al., 2019; Huang et al., 2020). In this paper, given a selective classifier, the result of selective risk is shown in the form of risk-coverage curves, which shows the selective risk of the selective classifier against its coverage. According to the object of selective classification, a selective classifier with a lower risk-coverage curve is better.

**Networks.** Following Huang et al. (2020); Xin et al. (2021), for image classification and text classification, we use VGG-16 (Simonyan & Zisserman, 2014) and BERT-base (Devlin et al., 2019) as the backbones of selective classifiers, respectively. More details of the backbone models and their training procedures are provided in Appendix C.2.

**Baselines.** We use SR (Geifman & El-Yaniv, 2017), SN (SelectiveNet) (Geifman & El-Yaniv, 2019), Gambler (Liu et al., 2019), SAT (Huang et al., 2020), and Reg-curr (Xin et al., 2021) for both image classification and text classification. Note that the SN is optimized for fixed coverage, so the comprehensive metrics AURC and risk-coverage curve, which summarizes performances under different coverages, are not suitable for evaluating the SN. Thus, we only evaluate the selective risk for a fixed coverage of the SN ensemble and provide the results in Appendix E. To ensure a fair comparison, for tasks of image classification, all the baselines are re-implemented based on the open resource code of SAT (Huang et al., 2020), and for tasks of text classification, they are re-implemented based on the open resource code of Reg-curr (Xin et al., 2021). The details of the hyperparameters of each baseline are provided in Appendix C.3.

### 6.1 VERIFICATION OF THE ASSUMPTIONS

We examine Assumption 1 and Assumption 2 (*only* for the baseline of SR) on datasets for both image classification and text classification. Since the definite samples are the idealization of samples on which member models provide almost the same predictive probability distributions, we take samples with a *standard deviation of predictive probability distributions of member models* (or STD for short) less than a small positive number $\epsilon$ as definite samples and the other samples as ambiguous samples

---

[1] Beyond Theorem 2, we provide an elaborate analysis on the lower bound of $\phi_0$ in Appendix G.

in experiments. Formally, $\text{STD} := \sqrt{\frac{\sum_{j=1}^{M}(\hat{\boldsymbol{\pi}}_j - \frac{1}{M}\sum_{i=1}^{M}\hat{\boldsymbol{\pi}}_i)^2}{M-1}}$, where $\hat{\boldsymbol{\pi}}_j$ is the predictive probability distribution vector of the $j$-th member model, and samples with $\text{STD} < \epsilon$ approximates definite samples in experiments. We choose $\epsilon = 10^{-3}$ for datasets of image classification and $\epsilon = 10^{-2}$ for datasets of text classification. Figure 1(a) and Figure 1(e) show the selective error rates (selective risks) of samples with $\text{STD} < 10^{-3}$ and samples with $\text{STD} \geq 10^{-3}$ given a range of confidence thresholds (which approximate $\Pr(Err|D, \text{C} \geq \tau)$ and $\Pr(Err|A, \text{C} \geq \tau)$) on the test set of each dataset. The results show that the selective risk of samples with $\text{STD} < 10^{-3}$ is lower than that of samples with $\text{STD} \geq 10^{-3}$ for all confidence thresholds near 1 on all datasets, which verifies Assumption 1. Figure 1(b)-1(d) and Figure 1(f)-1(h) shows the histogram of confidence scores of samples with $\text{STD} < \epsilon$ and that of other samples, which approximate $p_{\text{C}}(\kappa|D)$ and $p_{\text{C}}(\kappa|A)$, on the test set of each dataset. The results show that the number of samples with $\text{STD} < 10^{-3}$ is non-zero in the top bin on all datasets, which verifies Assumption 2. In summary, Assumption 1 and 2 hold on all datasets.

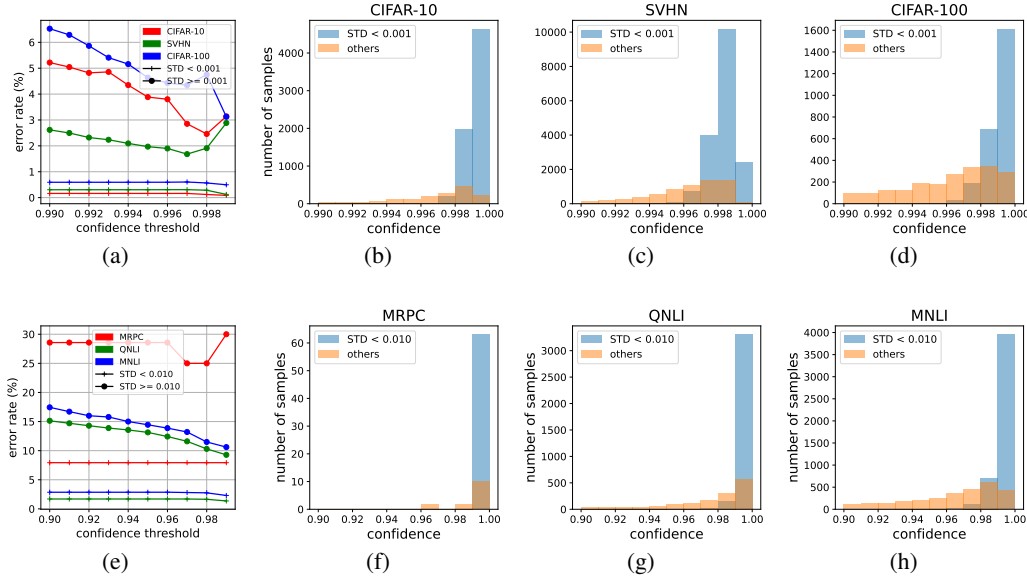

Figure 1: (a)/(e): the selective error rates (selective risks) of definite samples and ambiguous samples given a range of confidence thresholds on the test set of each dataset for image/text classification. (b)-(d): the histogram of confidence scores of samples with $\text{STD} < 10^{-3}$ and that of other samples on the test set of each dataset for image classification. (f)-(h): the histogram of confidence scores of samples with $\text{STD} < 10^{-2}$ and that of other samples on the test set of each dataset for text classification.

## 6.2 EVALUATION OF SELECTIVE CLASSIFIER ENSEMBLES

We first verify Theorem 2. Figure 2 shows the risk-coverage curves of the ensembles and the individual models of each baseline on each dataset. As we can see, except on MRPC, the risk-coverage curve of the ensemble is always entirely below that of the individual model, i.e., the ensemble has a lower selective risk than the individual model under any coverage, which is consistent with Theorem 2. The abnormal results on MRPC dataset may be because of the small number of samples in MRPC. The development set of MRPC has only 0.4k samples, which is much smaller than development sets of other datasets (see Table 2). More importantly, when the coverage is low, say 10%, only about 40 samples in MRPC are selected to predict, which may cause a large variance in selective risk estimation. Thus, the estimation of selective risk is not accurate under low coverage, which may explain the violation of Theorem 2 on MRPC. In summary, except the results on MRPC, which might have a large variance in selective risk estimation, the experimental results in Figure 2 verify the correctness and practicability of Theorem 2.

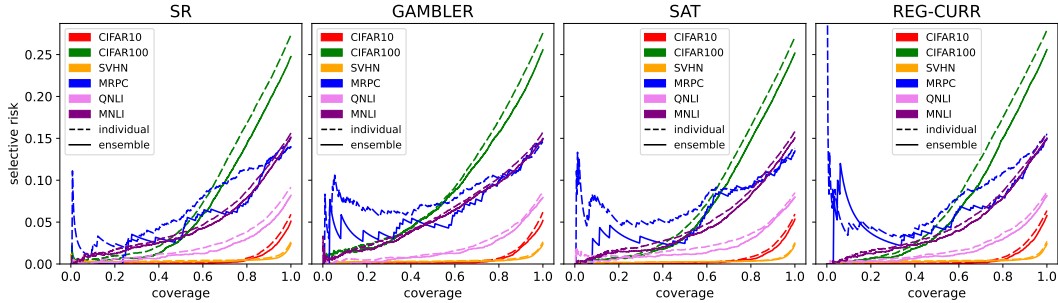

Figure 2: Risk-coverage curves of ensembles and individual models of each baseline on multiple datasets, where all ensembles consist of 5 member models.

Table 1: AURC/$10^{-4}$ on each dataset, where MNLI-(m) is the matched part of the MNLI development set. The means and standard deviations are calculated over three trials. The best entries are marked in bold.

| Dataset | #Member | SR | Gambler | SAT | Reg-curr |
|---|---|---|---|---|---|
| CIFAR-10 | 1 | **66.1±5.6** | $o=2.20$ 70.7±2.6 | 67.1±0.3 | 67.1±2.0 |
| | 2 | **55.5±1.7** | $o=2.20$ 63.2±1.8 | 63.0±0.4 | 59.2±1.1 |
| | 3 | **53.0±1.7** | $o=2.20$ 60.8±1.2 | 59.1±1.1 | 56.0±1.6 |
| | 4 | **51.3±2.2** | $o=2.20$ 58.9±2.0 | 58.3±0.5 | 54.6±1.0 |
| | 5 | **51.2±2.0** | $o=2.20$ 57.4±1.0 | 57.6±0.6 | 53.5±0.5 |
| CIFAR-100 | 1 | **793.1±9.0** | $o=4.60$ 930.6±10.2 | 807.3±6.0 | 851.9±12.8 |
| | 2 | **722.6±3.5** | $o=4.60$ 880.7±7.0 | 756.7±1.6 | 767.6±1.1 |
| | 3 | **695.9±2.6** | $o=4.60$ 871.3±4.3 | 741.2±3.8 | 739.5±5.7 |
| | 4 | **681.8±4.4** | $o=4.60$ 859.0±4.6 | 729.2±6.9 | 721.8±7.9 |
| | 5 | **672.5±1.9** | $o=4.60$ 857.1±4.7 | 715.5±12.2 | 714.2±6.3 |
| SVHN | 1 | 46.7±0.2 | $o=2.60$ 44.2±2.5 | **35.0±0.5** | 43.8±1.0 |
| | 2 | 42.1±1.8 | $o=2.60$ 41.3±2.9 | **33.5±0.8** | 39.9±1.4 |
| | 3 | 39.4±0.8 | $o=2.60$ 38.2±0.8 | **32.9±1.1** | 38.2±0.3 |
| | 4 | 38.1±0.9 | $o=2.60$ 37.2±1.0 | **32.7±0.9** | 37.0±1.4 |
| | 5 | 37.1±0.3 | $o=2.60$ 37.0±0.8 | **32.4±0.9** | 36.6±1.2 |
| MRPC | 1 | **654.3±68.3** | $o=1.80$ 695.6±65.9 | 794.6±51.9 | 696.8±66.0 |
| | 2 | **560.1±57.0** | $o=1.80$ 721.4±38.2 | 643.4±42.2 | 631.4±56.0 |
| | 3 | **566.0±57.3** | $o=1.80$ 707.3±58.4 | 581.2±11.0 | 653.7±67.3 |
| | 4 | **560.2±56.2** | $o=1.80$ 707.3±40.0 | 571.5±5.3 | 645.8±55.0 |
| | 5 | **561.3±55.6** | $o=1.80$ 650.1±24.8 | 563.0±16.5 | 640.3±14.1 |
| QNLI | 1 | 221.5 ±8.1 | $o=1.60$ 235.8 ±35.1 | 246.6 ±4.2 | **201.0 ±1.3** |
| | 2 | 196.4 ±7.0 | $o=1.60$ 195.5 ±12.8 | 220.0 ±1.1 | **186.9 ±1.5** |
| | 3 | 180.1 ±4.4 | $o=1.60$ 188.4 ±9.0 | 201.5 ±4.6 | **176.6 ±3.3** |
| | 4 | 175.7 ±3.5 | $o=1.60$ 181.0 ±6.2 | 196.0 ±5.5 | **174.3 ±3.5** |
| | 5 | 173.3 ±1.8 | $o=1.60$ 177.7 ±5.3 | 192.9 ±1.8 | **171.9 ±2.4** |
| MNLI-(m) | 1 | 515.3 ±8.6 | $o=2.80$ 607.6 ±19.3 | 519.8 ±8.0 | **496.8 ±9.5** |
| | 2 | 482.3 ±9.3 | $o=2.80$ 590.6 ±19.4 | 482.8 ±16.5 | **466.9 ±7.7** |
| | 3 | 475.0 ±8.4 | $o=2.80$ 563.0 ±11.4 | 470.1 ±6.5 | **458.5 ±5.3** |
| | 4 | 467.1 ±2.8 | $o=2.80$ 567.1 ±14.9 | 460.1 ±6.4 | **453.0 ±1.0** |
| | 5 | 465.2 ±3.8 | $o=2.80$ 569.4 ±11.4 | 454.4 ±3.1 | **451.9 ±2.1** |

Next, we show the AURCs of the ensembles in Table 1, where $o$ is the hyperparameter of Deep Gambler, and each ensemble has 2-5 member models. As we can see, the ensemble with five members has a significantly lower AURC than the individual model on each dataset. On CIFAR-10, CIFAR-100, SVHN, MRPC, QNLI, and MNLI, the AURCs of the best ensemble models with five members are 23%, 15%, 7%, 14%, 14%, and 9% lower than those of the best individual models. Among all ensembles, SR ensemble has the lowest AURCs on CIFAR-10, CIFAR-100, and MRPC,

SAT ensemble has the lowest AURCs on SVHN[2], and Reg-curr has the lowest AURCs on QNLI and MNLI[3].

Surprisingly, the SR ensemble and Reg-curr ensemble are state-of-the-art selective classifiers for image and text classification tasks, respectively. They only use the maximum probability as the confidence score, while SAT and SN use more sophisticated and explicit confidence score functions. Note that, on SVHN, the SAT ensemble performs better than the SR ensemble. We find that the annotations on SVHN are noisy, and after manually removing some noisy samples in the training set, the SR ensemble performs better than SAT ensemble. Therefore, the SVHN result indicates that SAT is better at handling label noise rather than selecting when to abstain. Please refer to Appendix F.3 for more details.

In addition, we conduct experiments to explore further properties of the selective classifier ensemble, including the effect of the number of members and the relationship between the classification performance and the selective classification performance (see Appendix F). The results are that an ensemble with more member models has better selective performance, and good classification performance of the ensemble does not necessarily imply good selective classification performance.

## 7    DISCUSSION

A possible direction for future work is to adapt our analysis to *standard classification*. The previous analyses of the randomization-based ensemble for standard classification need some impractical assumptions (Tumer & Ghosh, 1996; Fumera & Roli, 2005). On the contrary, this paper's assumptions are a good approximation of practical settings (see Section 6.1), and more importantly, the standard classification is a particular case of selective classification, i.e., selective classification with coverage of 1. Therefore, our analysis (although it does not cover the case of coverage of 1, i.e., the case of standard classification) may motivate the analysis of the randomization-based ensemble for standard classification in practical settings.

Another possible direction for future work is the relaxation of assumptions. This paper's assumption is a little strong for the convenience of proof. Although the experimental results suggest that these assumptions are the actual behaviors of the SR model, we guess the assumptions can be relaxed while the conclusion keeps the same. It is interesting to relax the assumption of the idealization of the definite samples. Although the idealization may be a good approximation of practical setting, it is unrealistic anyway. We believe that a similar theoretical result holds in the absence of the idealization.

## 8    CONCLUSION

We prove that under some assumptions, the ensemble has a lower selective risk than the individual model under a range of coverage. Although the metrics of selective classification are non-convex, we complete the proof with the help of several assumptions motivated by empirical observations. The assumptions and the result are well supported by the experimental results on multiple datasets of image classification tasks and text classification tasks. A surprising empirical result is that two simple methods, SR ensemble and its variant Reg-curr ensemble, (which can be summarized as *the ensemble models with maximum probability as confidence*) are state-of-the-art selective classifiers.

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

## A    RELATIONSHIP BETWEEN SELECTIVE CLASSIFICATION AND OOD DETECTION

Figure 3 shows the relationship between selective classification and OOD detection (or open set recognition, novelty detection).

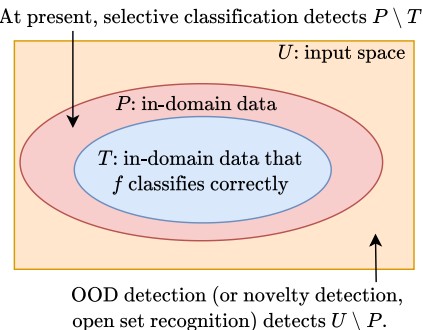

Figure 3: Relationship between selective classification and OOD detection (or open set recognition, novelty detection).

## B    PROOFS

The complete proof is somewhat complex, but its intuition is straightforward. In a word, the intuition is that because the ensemble avoids being overconfident over ambiguous samples, the ensemble has a lower selective risk. Details of the intuition are as follows:

1. the individual model is not "modest" over both ambiguous samples and definite samples (Assumption 2);

2. by contrast, based on Assumption 3, we prove that the ensemble provides modest confidence to ambiguous samples (Proposition 1). In addition, the confidence of definite samples remains the same throughout ensembling (due to the definition of definite samples);

3. thus, when confidence approaches 1, as long as the classifier's error rate over definite samples is lower than the error rate over ambiguous samples (Assumption 1), the individual model suffers more error predictions that come with the ambiguous samples than the ensemble. Based on this, we prove that the selective risk drops under a range of coverage via ensembling (Theorem 2).

We prove Proposition 1 in Section B.1 and prove Theorem 2 in Section B.2. The road map of the complete proof of the final result, i.e., Theorem 2, is shown in Figure 4.

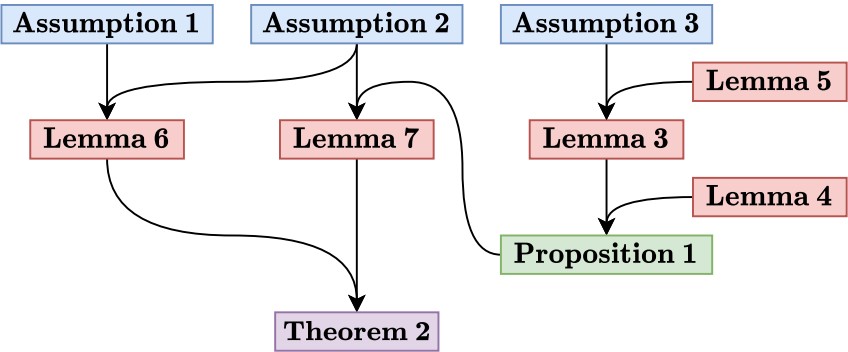

Figure 4: The road map of the proof of Theorem 2

### B.1 PROOF OF PROPOSITION 1

To prove Proposition 1, that is, the ensemble is *modest* over ambiguous samples, we first show that over ambiguous samples, the ensemble provides a moderate predictive probability for each class. By *moderate predictive probability*, we mean a predictive probability whose PDF approaches zero when itself approaches one or zero [4]. Formally, we have the following lemma (proof is provided in Section B.1.1).

**Lemma 3.** *If Assumption 3 holds, then*

$$p_{\Pi_{\text{ens}}^k}(0|A) = 0, \tag{10}$$

$$p_{\Pi_{\text{ens}}^k}(1|A) = 0, \tag{11}$$

*and*

$$p_{\Pi_{\text{ens}}^k}(\pi_{\text{ens}}^k|A) = O((\pi_{\text{ens}}^k)^{M-1}) \ (\pi_{\text{ens}}^k \to 0^+), \tag{12}$$

$$p_{\Pi_{\text{ens}}^k}(\pi_{\text{ens}}^k|A) = O((1 - \pi_{\text{ens}}^k)^{M-1}) \ (\pi_{\text{ens}}^k \to 1^-), \tag{13}$$

*where the notation follows that of Assumption 3, and $\Pi_{\text{ens}}^k$ is the predictive probability of the ensemble for class $k$.*

Secondly, to prove the ensemble is modest over ambiguous samples, we show the relationship between the PDF of confidence and the PDFs of predictive probabilities. Note that the confidence of an SR model is the maximum predictive probability. Thus, the following lemma (proof is provided in Section B.1.2) bounds the PDF of confidence by PDFs of predictive probabilities, and then it is clear that the ensemble is modest over ambiguous samples, considering the moderate predictive probabilities of the ensemble over ambiguous samples.

---

[4]Thus, Proposition 1, or that the ensemble is modest over ambiguous samples, is equivalent to that the ensemble provides moderate confidence over ambiguous samples

**Lemma 4.** *Let $\Pi^k$ ($1 \le k \le K$) be $K$ continuous random variables, and $C := \max_k \Pi^k$. Then we have*

$$p_C(\kappa) \le \sum_{k=1}^{K} p_{\Pi^k}(\kappa). \tag{14}$$

Finally, since $C_{\mathrm{ens}} = \max_k \Pi_{\mathrm{ens}}^k$, Lemma 3 and Lemma 4 derive that when $\kappa_{\mathrm{ens}} \to 1^-$,

$$p_{C_{\mathrm{ens}}}(\kappa_{\mathrm{ens}}|A) \le \sum_{k=1}^{K} p_{\Pi_{\mathrm{ens}}^k}(\kappa_{\mathrm{ens}}|A) = O((1 - \kappa_{\mathrm{ens}})^{M-1}),$$

and thus Proposition 1 holds.

### B.1.1 PROOF OF LEMMA 3

We first derive the PDF of the average of multiple continuous random variables in terms of the PDFs of these random variables (Lemma 5), which helps us to analyze the PDF of the ensemble's predictive probabilities.

**Lemma 5.** *Let $X_1, X_2, \ldots, X_M$ be $M$ continuous random variables, and their average is $X_{\mathrm{avg}} := \frac{1}{M}\sum_{i=1}^{M} X_i$. Then the PDF of $X_{\mathrm{avg}}$ is*

$$p_{X_{\mathrm{avg}}}(x_{\mathrm{avg}}) = M \int_{\mathbb{R}^{M-1}} \mathrm{d}x_1 \mathrm{d}x_2 \cdots \mathrm{d}x_{M-1} \cdot p_{\vec{X}}(x_1, x_2, \ldots, x_{M-1}, Mx_{\mathrm{avg}} - \sum_{i=1}^{M-1} x_i), \tag{15}$$

*where $p_{\vec{X}}$ is $p_{X_1, X_2, \ldots, X_M}$ for short.*

*Proof.* The distribution function of $X_{\mathrm{avg}}$ is

$$
\begin{aligned}
F_{X_{\mathrm{avg}}}(x_{\mathrm{avg}}) &= \int_{\sum_i x_i \le Mx_{\mathrm{avg}}} \mathrm{d}x_1 \cdots \mathrm{d}x_{M-1} \mathrm{d}x_M \cdot p_{\vec{X}}(x_1, \ldots, x_M) \\
&= \int_{\mathbb{R}^{M-1}} \mathrm{d}x_1 \cdots \mathrm{d}x_{M-1} \int_{-\infty}^{Mx_{\mathrm{avg}} - \sum_{i=1}^{M-1} x_i} \mathrm{d}x_M \cdot p_{\vec{X}}(x_1, \cdots, x_M).
\end{aligned}
$$

Let $x_M = u - \sum_{i=1}^{M-1} x_i$, then the integral above is equal to

$$
\begin{aligned}
&\int_{\mathbb{R}^{M-1}} \mathrm{d}x_1 \cdots \mathrm{d}x_{M-1} \int_{-\infty}^{Mx_{\mathrm{avg}}} \mathrm{d}u \cdot p_{\vec{X}}(x_1, \ldots, x_{M-1}, u - \sum_{i=1}^{M-1} x_i) \\
&= \int_{-\infty}^{Mx_{\mathrm{avg}}} \mathrm{d}u \int_{\mathbb{R}^{M-1}} \mathrm{d}x_1 \cdots \mathrm{d}x_{M-1} \cdot p_{\vec{X}}(x_1, \ldots, x_{M-1}, u - \sum_{i=1}^{M-1} x_i). \tag{16}
\end{aligned}
$$

The PDF of $X_{\mathrm{avg}}$ is the derivative of $F_{X_{\mathrm{avg}}}$, which, combined with (,16) derives

$$
\begin{aligned}
p_{X_{\mathrm{avg}}}(x_{\mathrm{avg}}) &= F'_{X_{\mathrm{avg}}}(x_{\mathrm{avg}}) \\
&= \frac{\mathrm{d}(Mx_{\mathrm{avg}})}{\mathrm{d}x_{\mathrm{avg}}} \cdot \frac{\mathrm{d}F_{X_{\mathrm{avg}}}}{\mathrm{d}(Mx_{\mathrm{avg}})} \\
&= M \int_{\mathbb{R}^{M-1}} \mathrm{d}x_1 \cdots \mathrm{d}x_{M-1} \cdot p_{\vec{X}}(x_1, \ldots, x_{M-1}, Mx_{\mathrm{avg}} - \sum_{i=1}^{M-1} x_i),
\end{aligned}
$$

which is exactly (15). $\qquad\square$

**Proof of Lemma 3.** Based on Lemma 5 and Assumption 3, we prove Lemma 3 as follows.

*Proof.* With Lemma 5 applied to $\Pi_i^k$, $1 \le i \le M$, we have

$$p_{\Pi_{\mathrm{ens}}^k}(\pi_{\mathrm{ens}}^k|A) = M \int_{\mathbb{R}^{M-1}} \mathrm{d}\pi_1^k \cdots \mathrm{d}\pi_{M-1}^k \cdot p_{\vec{\Pi}^k}(\pi_1^k, \ldots, \pi_{M-1}^k, M\pi_{\mathrm{ens}}^k - \sum_{i=1}^{M-1} \pi_i^k|A), \tag{17}$$

where $p_{\vec{\Pi}^k}$ is $p_{\Pi_1^k, \dots, \Pi_M^k}$ for short. The integrand in the right-hand side of (17) being non-zero requires

$$\begin{cases} 0 \le \pi_i^k \le 1, i = 1, 2, \dots, M-1 \\ 0 \le M\pi_{\text{ens}}^k - \sum_{i=1}^{M-1} \pi_i^k \le 1 \end{cases} . \tag{18}$$

Firstly, we prove (10) and (12). When $0 \le \pi_{\text{ens}}^k \le \frac{1}{M}$, it easy to verify that (18) is equivalent to

$$\begin{cases} 0 \le \pi_1^k \le M\pi_{\text{ens}}^k \\ 0 \le \pi_2^k \le M\pi_{\text{ens}}^k - \pi_1^k \\ \cdots \\ 0 \le \pi_i^k \le M\pi_{\text{ens}}^k - \pi_1^k - \cdots - \pi_{i-1}^k \\ \cdots \\ 0 \le \pi_{M-1}^k \le M\pi_{\text{ens}}^k - \pi_1^k - \cdots - \pi_{M-2}^k \end{cases} . \tag{19}$$

Thus, (17) transforms into

$$p_{\Pi_{\text{ens}}^k}(\pi_{\text{ens}}^k|A) = M \int_0^{M\pi_{\text{ens}}^k} d\pi_1^k \cdots \int_0^{M\pi_{\text{ens}}^k - \sum_{j=1}^{i-1} \pi_j^k} d\pi_i^k \cdots \int_0^{M\pi_{\text{ens}}^k - \sum_{j=1}^{M-2} \pi_j^k} d\pi_{M-1}^k$$
$$\cdot p_{\vec{\Pi}^k}(\pi_1^k, \dots, \pi_{M-1}^k, M\pi_{\text{ens}}^k - \sum_{i=1}^{M-1} \pi_i^k | A).$$

Considering that $p_{\vec{\Pi}^k}(\cdot|A)$ is bounded, as Assumption 3 claims, let $B$ be one of its upper bounds. Then we have

$$p_{\Pi_{\text{ens}}^k}(\pi_{\text{ens}}^k|A) \le M \int_0^{M\pi_{\text{ens}}^k} d\pi_1^k \cdots \int_0^{M\pi_{\text{ens}}^k - \sum_{j=1}^{i-1} \pi_j^k} d\pi_i^k \cdots \int_0^{M\pi_{\text{ens}}^k - \sum_{j=1}^{M-2} \pi_j^k} d\pi_{M-1}^k B$$
$$\le M \int_0^{M\pi_{\text{ens}}^k} d\pi_1^k \cdots \int_0^{M\pi_{\text{ens}}^k} d\pi_i^k \cdots \int_0^{M\pi_{\text{ens}}^k} d\pi_{M-1}^k B$$
$$= MB \cdot \int_0^{M\pi_{\text{ens}}^k} d\pi_1^k \cdots \int_0^{M\pi_{\text{ens}}^k} d\pi_{M-1}^k$$
$$= MB \cdot (M\pi_{\text{ens}}^k)^{M-1}$$
$$= M^M B \cdot (\pi_{\text{ens}}^k)^{M-1}, \tag{20}$$

which directly derives (10) and (12) (note that the PDF is non-negative).

Secondly, we prove (11) and (13). These two equations could be derived like (10) and (12). However, here we take another way that uses a little trick to simplify the proof. Let $\Pi_i^k$ and $\Pi_{\text{ens}}^k$ be the corresponding random variables of $\pi_i^k$ and $\pi_{\text{ens}}^k$ respectively, and $U_i = 1 - \Pi_i^k$, $U_{\text{ens}} = \frac{1}{M} \sum_{i=1}^M U_i = 1 - \Pi_{\text{ens}}^k$. Applying (20) to $U_i$ and $U_{\text{ens}}$, we get that when $0 \le u_{\text{ens}} \le \frac{1}{M}$,

$$p_{U_{\text{ens}}}(u_{\text{ens}}|A) \le M^M B \cdot u_{\text{ens}}^{M-1}. \tag{21}$$

It is easy to see that $p_{U_{\text{ens}}}(u_{\text{ens}}|A) = p_{\Pi_{\text{ens}}^k}(1 - u_{\text{ens}}|A)$. Sticking this into (21), we have

$$p_{\Pi_{\text{ens}}^k}(1 - u_{\text{ens}}|A) \le M^M B \cdot u_{\text{ens}}^{M-1},$$

when $0 \le u_{\text{ens}} \le \frac{1}{M}$. With the $(1 - u_{\text{ens}})$ in the equation above replaced with $\pi_{\text{ens}}^k$, we have

$$p_{\Pi_{\text{ens}}^k}(\pi_{\text{ens}}^k|A) \le M^M B \cdot (1 - \pi_{\text{ens}}^k)^{M-1},$$

when $1 - \frac{1}{M} \le \pi_{\text{ens}}^k \le 1$, which directly derives (11) and (13). $\square$

### B.1.2 PROOF OF LEMMA 4

*Proof.* First of all, we prove $\forall \kappa_1, \kappa_2, \kappa_1 < \kappa_2$,

$$F_C(\kappa_2) - F_C(\kappa_1) \le \sum_{k=1}^K F_{\Pi^k}(\kappa_2) - F_{\Pi^k}(\kappa_1) \tag{22}$$

It is easy to see that

$$F_{\text{C}}(\kappa) = F_{\Pi^1,\dots,\Pi^K}(\kappa,\dots,\kappa) = \int_{(-\infty,\kappa]^K} \mathrm{d}\pi^1 \cdots \mathrm{d}\pi^K \, p_{\Pi^1,\dots,\Pi^K}(\pi^1,\cdots,\pi^K),$$

so the left-hand side of (22) is

$$\int_{(-\infty,\kappa_2]^K} \mathrm{d}\pi^1 \cdots \mathrm{d}\pi^K \, p_{\Pi^1,\dots,\Pi^K}(\pi^1,\cdots,\pi^K)$$

$$- \int_{(-\infty,\kappa_1]^K} \mathrm{d}\pi^1 \cdots \mathrm{d}\pi^K \, p_{\Pi^1,\dots,\Pi^K}(\pi^1,\cdots,\pi^K)$$

$$= \int_{(-\infty,\kappa_2]^K \setminus (-\infty,\kappa_1]^K} \mathrm{d}\pi^1 \cdots \mathrm{d}\pi^K \, p_{\Pi^1,\dots,\Pi^K}(\pi^1,\cdots,\pi^K), \tag{23}$$

where the last equality is due to $(-\infty,\kappa_1] \subset (-\infty,\kappa_2]$, and the right-hand side of (22) is

$$\sum_{k=1}^{K} \int_{[\kappa_1,\kappa_2]} \mathrm{d}\pi^k \, p_{\Pi^k}(\pi^k)$$

$$= \sum_{k=1}^{K} \int_{\mathbb{R}^{k-1} \times [\kappa_1,\kappa_2] \times \mathbb{R}^{K-k}} \mathrm{d}\pi^1 \cdots \mathrm{d}\pi^K \cdot p_{\Pi^1,\dots,\Pi^K}(\pi^1,\cdots,\pi^K)$$

$$\geq \int_{\bigcup_{k=1}^{K} \mathbb{R}^{k-1} \times [\kappa_1,\kappa_2] \times \mathbb{R}^{K-k}} \mathrm{d}\pi^1 \cdots \mathrm{d}\pi^K \cdot p_{\Pi^1,\dots,\Pi^K}(\pi^1,\cdots,\pi^K), \tag{24}$$

where the last inequality is because $\mathbb{R}^{k-1} \times [\kappa_1,\kappa_2] \times \mathbb{R}^{K-k}$ for different $k$, $1 \leq k \leq K$, may have an intersection. To prove (22), we only need to prove that the right-hand side of (23) is less than or equal to the right-hand side of (24), which is equivalent to prove

$$(-\infty,\kappa_2]^K \setminus (-\infty,\kappa_1]^K \subset \bigcup_{k=1}^{K} \mathbb{R}^{k-1} \times [\kappa_1,\kappa_2] \times \mathbb{R}^{K-k}. \tag{25}$$

Now we prove (25). $\forall (\pi^1,\dots,\pi^K) \in (-\infty,\kappa_2]^K \setminus (-\infty,\kappa_1]^K$, we have

$$\forall k, 1 \leq k \leq K, \pi^k \leq \kappa_2, \tag{26}$$

$$\exists k_0, 1 \leq k_0 \leq K, \pi^{k_0} > \kappa_1, \tag{27}$$

where (27) is because if all $\pi^k$ is less than or equal to $\kappa_1$ instead, then $(\pi^1,\dots,\pi^K) \in (-\infty,\kappa_1]^K$, which contradicts with $(\pi^1,\dots,\pi^K) \in (-\infty,\kappa_2]^K \setminus (-\infty,\kappa_1]^K$. Thus, $\pi^{k_0} \in [\kappa_1,\kappa_2]$, so

$$(\pi^1,\dots,\pi^K) \in \mathbb{R}^{k_0-1} \times [\kappa_1,\kappa_2] \times \mathbb{R}^{K-k_0} \subset \bigcup_{k=1}^{K} \mathbb{R}^{k-1} \times [\kappa_1,\kappa_2] \times \mathbb{R}^{K-k},$$

which is precisely (25), and therefore (22) is proved.

With (22) and the definition of derivatives, it is easy to see that $F_{\text{C}}'(\kappa) \leq \sum_{k=1}^{K} F_{\Pi^k}'(\kappa)$, which is equivalent to $p_{\text{C}}(\kappa) \leq \sum_{k=1}^{K} p_{\Pi^k}(\kappa)$. Thus, Lemma 4 is proved. □

### B.2 PROOF OF THEOREM 2

The intuition of the proof is as follows. Intuitively, since the ensemble is more modest than the individual over ambiguous samples and is the same modest as the individual model over definite samples, the ensemble tends to select more definite samples when the confidence threshold approaches 1, compared with the individual model (Lemma 7). Thus, (still intuitively) as long as the selective risk over definite samples is lower than that over ambiguous samples when the confidence threshold approaches 1 (Assumption 1), the ensemble is certainly to have a lower selective risk when the confidence threshold approaches 1.

Although the intuition is straightforward, the rigorous proof is not easy. For the convenience of the proof, we show Lemma 6 and Lemma 7 first. Lemma 6 claims that for the individual model, the selective risk given definite samples is lower than the overall selective risk. Lemma 7 claims that the ensemble is unlikely to select ambiguous samples to predict when the confidence threshold approaches 1.

**Lemma 6.** *If Assumption 1-2 holds, then for any individual SR model,*

$$\lim_{\tau \to 1^-} R(\phi(\tau)) > \lim_{\tau \to 1^-} \Pr(Err|D, \mathbf{C} \geq \tau), \tag{28}$$

*where the notation follows those of Assumption 1-2.*

*Proof.* Using Bayes' rule, we have

$$\Pr(A|\mathbf{C} \geq \tau) = \frac{\Pr(\mathbf{C} \geq \tau|A)\Pr(A)}{\Pr(\mathbf{C} \geq \tau|A)\Pr(A) + \Pr(\mathbf{C} \geq \tau|D)\Pr(D)}. \tag{29}$$

Using L'Hospital's rule, we have

$$
\begin{aligned}
\lim_{\tau \to 1^-} \frac{\Pr(\mathbf{C} \geq \tau|D)}{\Pr(\mathbf{C} \geq \tau|A)} &= \lim_{\tau \to 1^-} \frac{\int_\tau^1 p_{\mathbf{C}}(\kappa|D)\mathrm{d}\kappa}{\int_\tau^1 p_{\mathbf{C}}(\kappa|A)\mathrm{d}\kappa} \\
&= \lim_{\tau \to 1^-} \frac{-p_{\mathbf{C}}(\tau|D)}{-p_{\mathbf{C}}(\tau|A)} \\
&= \lim_{\tau \to 1^-} \frac{p_{\mathbf{C}}(\tau|D)}{p_{\mathbf{C}}(\tau|A)} \\
&> 0,
\end{aligned} \tag{30}
$$

where the last inequality is due to Assumption 2. Combining this with (29), we have

$$
\begin{aligned}
\lim_{\tau \to 1^-} \Pr(A|\mathbf{C} \geq \tau) &= \frac{\Pr(A)}{\Pr(A) + \Pr(D)\lim_{\tau \to 1^-} \frac{\Pr(\mathbf{C} \geq \tau|D)}{\Pr(\mathbf{C} \geq \tau|A)}}, \\
&= \frac{\Pr(A)}{\Pr(A) + \Pr(D)\lim_{\tau \to 1^-} \frac{p_{\mathbf{C}}(\tau|D)}{p_{\mathbf{C}}(\tau|A)}}, \\
&> 0.
\end{aligned} \tag{31}
$$

Now we derive (28).

$$
\begin{aligned}
R(\phi(\tau)) &= \Pr(Err, A|\mathbf{C} \geq \tau) + \Pr(Err, D|\mathbf{C} \geq \tau) \\
&= \Pr(Err|A, \mathbf{C} \geq \tau)\Pr(A|\mathbf{C} \geq \tau) + \Pr(Err|D, \mathbf{C} \geq \tau)\Pr(D|\mathbf{C} \geq \tau) \\
&= \Pr(Err|A, \mathbf{C} \geq \tau)\Pr(A|\mathbf{C} \geq \tau) + \Pr(Err|D, \mathbf{C} \geq \tau)[1 - \Pr(A|\mathbf{C} \geq \tau)] \\
&= [\Pr(Err|A, \mathbf{C} \geq \tau) - \Pr(Err|D, \mathbf{C} \geq \tau)]\Pr(A|\mathbf{C} \geq \tau) + \Pr(Err|D, \mathbf{C} \geq \tau).
\end{aligned}
$$

According to the equation above, we have

$$
\begin{aligned}
\lim_{\tau \to 1^-} R(\phi(\tau)) &= [\lim_{\tau \to 1^-} \Pr(Err|A, \mathbf{C} \geq \tau) - \lim_{\tau \to 1^-} \Pr(Err|D, \mathbf{C} \geq \tau)] \\
&\quad \cdot \lim_{\tau \to 1^-} \Pr(A|\mathbf{C} \geq \tau) + \lim_{\tau \to 1^-} \Pr(Err|D, \mathbf{C} \geq \tau)
\end{aligned} \tag{32}
$$

Due to (31) and Assumption 1, the first term of the equation above is positive, so (28) is derived. $\square$

**Lemma 7.** *If Assumption 2-3 hold, then*

$$\lim_{\tau_{\mathrm{ens}} \to 1^-} \Pr(A|\mathbf{C}_{\mathrm{ens}} \geq \tau_{\mathrm{ens}}) = 0, \tag{33}$$

*where the notation follows those of Assumption 2-3.*

*Proof.* Similar to (31), with Bayes' rule and L'Hospital's rule, we can derive that

$$
\begin{aligned}
\lim_{\tau_{\mathrm{ens}} \to 1^-} \Pr(A|\mathbf{C}_{\mathrm{ens}} \geq \tau_{\mathrm{ens}}) &= \frac{\Pr(A)\lim_{\tau_{\mathrm{ens}} \to 1^-} \frac{\Pr(\mathbf{C}_{\mathrm{ens}} \geq \tau_{\mathrm{ens}}|A)}{\Pr(\mathbf{C}_{\mathrm{ens}} \geq \tau_{\mathrm{ens}}|D)}}{\Pr(A)\lim_{\tau_{\mathrm{ens}} \to 1^-} \frac{\Pr(\mathbf{C}_{\mathrm{ens}} \geq \tau_{\mathrm{ens}}|A)}{\Pr(\mathbf{C}_{\mathrm{ens}} \geq \tau_{\mathrm{ens}}|D)} + \Pr(D)}, \\
&= \frac{\Pr(A)\lim_{\tau_{\mathrm{ens}} \to 1^-} \frac{p_{\mathbf{C}_{\mathrm{ens}}}(\tau_{\mathrm{ens}}|A)}{p_{\mathbf{C}_{\mathrm{ens}}}(\tau_{\mathrm{ens}}|D)}}{\Pr(A)\lim_{\tau_{\mathrm{ens}} \to 1^-} \frac{p_{\mathbf{C}_{\mathrm{ens}}}(\tau_{\mathrm{ens}}|A)}{p_{\mathbf{C}_{\mathrm{ens}}}(\tau_{\mathrm{ens}}|D)} + \Pr(D)},
\end{aligned} \tag{34}
$$

where $C_{ens}$ is the confidence score of the ensemble. Because for a definite sample, the confidence score of the ensemble is equal to that of the individual model, we have

$$\lim_{\tau_{ens}\to1^-}\frac{p_{C_{ens}}(\tau_{ens}|A)}{p_{C_{ens}}(\tau_{ens}|D)} = \lim_{\tau_{ens}\to1^-}\frac{p_{C_{ens}}(\tau_{ens}|A)}{p_C(\tau_{ens}|D)} = 0,$$

where the last equality is due to Proposition 1 and Assumption 2. Substituting this to (34), we obtain (33). $\qquad\square$

**Proof of Theorem 2.**

*Proof.* First, for the convenience of the proof, given an SR model $(f, g)$, we define a threshold-to-coverage function $\rho_{(f,g)}$ of $(f, g)$ that maps the confidence threshold to the corresponding coverage,

$$\rho_{(f,g)} : (0, 1) \to (0, 1), \tau \mapsto \phi(f, g; \tau).$$

Second, we prove that $\exists \delta \in (0, 1), \forall \tau_{ens} \in (1 - \delta, 1)$,

$$\Pr(A|C_{ens} \ge \tau_{ens}) - \Pr(Err_{ind}|C \ge \tau) + \Pr(Err_{ens}|D, C_{ens} \ge \tau_{ens}) < 0, \qquad (35)$$

$$\tau = \max \rho^{-1} \circ \rho_{ens}(\tau_{ens}) \qquad (36)$$

where C is the confidence score of the individual SR model, $Err_{ind}$ is the event that the individual model makes an error prediction, $Err_{ens}$ is the event that the ensemble makes an error prediction, $\rho$ and $\rho_{ens}$ are the threshold-to-coverage functions of the individual model and the ensemble respectively. Note that the symbol $\rho^{-1}$ denotes the preimage under $\rho$, rather than the inverse function of $\rho$. Because when $\tau_{ens} \to 1^-$, the coverage of the ensemble $\rho_{ens}(\tau_{ens})$ approaches 0, and the coverage of the individual model is equal to the coverage of the ensemble, i.e., $\rho(\tau) = \rho_{ens}(\tau_{ens})$, we have $\tau \to 1^-$ when $\tau_{ens} \to 1^-$. Thus,

$$\lim_{\tau_{ens}\to1^-} \Pr(Err_{ind}|C \ge \tau) = \lim_{\tau\to1^-} \Pr(Err_{ind}|C \ge \tau). \qquad (37)$$

In addition, for definite samples, the confidence score of the ensemble and that of the individual model are the same, and the ensemble and the individual model make error predictions on the same set of samples, i.e. $Err_{ind} = Err_{ens}$, so

$$\lim_{\tau_{ens}\to1^-} \Pr(Err_{ens}|D, C_{ens} \ge \tau_{ens}) = \lim_{\tau_{ens}\to1^-} \Pr(Err_{ens}|D, C \ge \tau_{ens})$$

$$= \lim_{\tau\to1^-} \Pr(Err_{ens}|D, C \ge \tau)$$

$$= \lim_{\tau\to1^-} \Pr(Err_{ind}|D, C \ge \tau), \qquad (38)$$

where the second equality is just a variable substitution. Finally, we have

$$\lim_{\tau_{ens}\to1^-} [\Pr(A|C_{ens} \ge \tau_{ens}) - \Pr(Err_{ind}|C \ge \tau) + \Pr(Err_{ens}|D, C_{ens} \ge \tau_{ens})]$$

$$= \lim_{\tau_{ens}\to1^-} [0 - \Pr(Err_{ind}|C \ge \tau) + \Pr(Err_{ens}|D, C_{ens} \ge \tau_{ens})]$$

$$= -\lim_{\tau_{ens}\to1^-} \Pr(Err_{ind}|C \ge \tau) + \lim_{\tau_{ens}\to1^-} \Pr(Err_{ind}|D, C \ge \tau)$$

$$= -\lim_{\tau\to1^-} \Pr(Err_{ind}|C \ge \tau) + \lim_{\tau\to1^-} \Pr(Err_{ind}|D, C \ge \tau)$$

$$< 0, \qquad (39)$$

where the first equality is due to Lemma 7, the second equality is due to (37) and (38), and the last inequality is due to Lemma 6. Thus, with (39), it is easy to see that (35) holds.

Third, we have

$$\Pr(Err_{ens}|C_{ens} \ge \tau_{ens}) = \Pr(Err_{ens}, A|C_{ens} \ge \tau_{ens}) + \Pr(Err_{ens}, D|C_{ens} \ge \tau_{ens})$$

$$= \Pr(Err_{ens}|A, C_{ens} \ge \tau_{ens})\Pr(A|C_{ens} \ge \tau_{ens})$$

$$\quad + \Pr(Err_{ens}|D, C_{ens} \ge \tau_{ens})\Pr(D|C_{ens} \ge \tau_{ens})$$

$$= \Pr(Err_{ens}|A, C_{ens} \ge \tau_{ens})\Pr(A|C_{ens} \ge \tau_{ens})$$

$$\quad + \Pr(Err_{ens}|D, C_{ens} \ge \tau_{ens})[1 - \Pr(A|C_{ens} \ge \tau_{ens})]$$

$$= [\Pr(Err_{ens}|A, C_{ens} \ge \tau_{ens}) - \Pr(Err_{ens}|D, C_{ens} \ge \tau_{ens})]$$

$$\quad \cdot \Pr(A|C_{ens} \ge \tau_{ens}) + \Pr(Err_{ens}|D, C_{ens} \ge \tau_{ens})$$

$$\le \Pr(A|C_{ens} \ge \tau_{ens}) + \Pr(Err_{ens}|D, C_{ens} \ge \tau_{ens}), \qquad (40)$$

where the last inequality is because any probability is in [0, 1]. Combining this with (35), we have $\exists \delta \in (0,1), \forall \tau_{\text{ens}} \in (1-\delta, 1)$,

$$
\begin{aligned}
\Pr(Err_{\text{ens}}|\mathrm{C}_{\text{ens}} \geq \tau_{\text{ens}}) \leq & \Pr(A|\mathrm{C}_{\text{ens}} \geq \tau_{\text{ens}}) + \Pr(Err_{\text{ens}}|D, \mathrm{C}_{\text{ens}} \geq \tau_{\text{ens}}) \\
< & \Pr(Err_{\text{ind}}|\mathrm{C} \geq \tau) - \Pr(Err_{\text{ens}}|D, \mathrm{C}_{\text{ens}} \geq \tau_{\text{ens}}) \\
& + \Pr(Err_{\text{ens}}|D, \mathrm{C}_{\text{ens}} \geq \tau_{\text{ens}}) \\
= & \Pr(Err_{\text{ind}}|\mathrm{C} \geq \tau),
\end{aligned} \tag{41}
$$

where the second inequality is due to (35). $\Pr(Err_{\text{ens}}|\mathrm{C}_{\text{ens}} \geq \tau_{\text{ens}})$ is the selective risk of the ensemble given the confidence threshold of $\tau_{\text{ens}}$ (or given coverage of $\rho_{\text{ens}}(\tau_{\text{ens}})$), and $\Pr(Err_{\text{ind}}|\mathrm{C} \geq \tau)$ is the selective risk of the individual model given the confidence threshold of $\tau$ (or given coverage of $\rho(\tau) = \rho_{\text{ens}}(\tau_{\text{ens}})$). These two selective risks are under the same coverage $\rho_{\text{ens}}(\tau_{\text{ens}})$. Thus, (41) is equivalent to that $\exists \delta \in (0,1), \forall \tau_{\text{ens}} \in (1-\delta, 1)$, the ensemble has a lower selective risk than the individual model given the coverage of $\phi = \rho_{\text{ens}}(\tau_{\text{ens}})$. This statement can be simplified as $\exists \delta \in (0,1), \forall \phi \in (0, \rho_{\text{ens}}(1-\delta))$, the ensemble has a lower selective risk than the individual model, given the coverage of $\phi$. $\square$

## C  DETAILS OF EXPERIMENTS

### C.1  DATASETS

The experiments were conducted on multiple data sets of image classification and text classification. The image classification datasets are CIFAR-10, CIFAR-100, (Krizhevsky, 2009) and SVHN (Netzer et al., 2011), whose image sizes are all $32 \times 32 \times 3$ pixels. The datasets of text classification are MRPC (Dolan & Brockett, 2005), MNLI (Williams et al., 2018) and QNLI (Wang et al., 2018). The task of MRPC is to judge whether two paragraphs of text are semantically equivalent. MNLI's task is to judge the inferential relationship between sentences (three categories). The task of QNLI is to determine whether a paragraph has the answer to a given question. The sizes of the training set, development set, and test set of each data set used in experiments are shown in Table 2. MNLI's development set and test set are divided into *matched* and *mismatched* parts. In the table, (m) represents matched, and (mm) represents mismatched. The matched parts are sampled from the same source as the training set, while the mismatched parts are sampled from different sources. Current selective classification only considers test samples from the same distribution as the training set, so only the matched parts are used in experiments. In addition, test sets of MRPC, QNLI, and MNLI are not accessible, so we use their development sets as test sets. According to Liu et al. (2019); Huang et al. (2020), since CIFAR-10, CIFAR-100 and SVHN originally had no development set, their development sets were 2000 samples randomly divided from corresponding test sets.

Table 2: Sizes of training sets, development sets, and test sets for each dataset used in experiments

| Datasets | Training Set | Development Set | Test Set | Number of Classes |
|---|---|---|---|---|
| CIFAR-10 | 50.0k | | 10.0k | 10 |
| CIFAR-100 | 50.0k | | 10.0k | 100 |
| SVHN | 73.3k | | 26.0k | 10 |
| MRPC | 3.7k | 0.4k | 1.7k | 2 |
| QNLI | 104.7k | 5.5k | 5.5k | 2 |
| MNLI | 392.7k | 9.8k (m)/ 9.8k(mm) | 9.8k(m)/9.8k(mm) | 3 |

### C.2  MODEL IMPLEMENTATIONS AND TRAINING PROCEDURES

For image classification, the backbone model is VGG-16 (Simonyan & Zisserman, 2014) with Dropout (Srivastava et al., 2014), batch normalization (Ioffe & Szegedy, 2015). It is trained in the same way as Huang et al. (2020). The model is optimized using SGD with an initial learning rate of 0.1 (the learning rate decays by half in every 25 epochs), the momentum of 0.9, weight decay of 0.0005, batch size of 128, and a total training epoch of 300. Data preprocessing includes

data augmentation (random cropping and flip) and normalization. The implementations of the backbone model and data preprocessing are based on the official open-sourced implementation of SAT to ensure a fair comparison.

For text classification, the backbone model of selective classifiers is BERT-base (Devlin et al., 2019). Pretrained BERT-base is provided by the Huggingface Transformer Library (Wolf et al., 2020). It is trained/fine-tuned in the same way as Xin et al. (2021), except on dataset MRPC. On QNLI and MNLI, the model is trained/fine-tuned using AdamW (Loshchilov & Hutter, 2017) for 3 epochs, with a learning rate of $2 \times 10^{-5}$, batch size of 32, and the maximum input sequence length of 128. On MRPC, the model is trained/fine-tuned for 10 epoch, with other settings the same as those on QNLI and MNLI. This unique setting of training epoch is due to the small number of samples in MRPC, which makes the training require more epochs to reach convergence on MRPC.

### C.3 HYPERPARAMETERS OF SELECTIVE CLASSIFIERS

For the hyperparameter $c$ of SN, we choose $c = 0.9$ for evaluating its selective risk, given the coverage of 90%. The results of SN are reported in Appendix E. For the hyperparameter $o$ of Gambler, we tune $o$ on validation sets in the same way as Liu et al. (2019). For the hyperparameter $\alpha$ of SAT, we set $\alpha = 0.99$, the same as Huang et al. (2020). For the hyperparameter $\lambda$ of Reg-curr, we set $\lambda = 0.05$.

## D  ADDITIONAL EXPERIMENTAL RESULTS

Table 3 and 4 shows the selective risks of ensembles under coverage 10%-100% on each dataset, where hyperparameters of Gambler are the same as those in Table 1. Notably, no ensemble consistently outperforms others under all coverage on all datasets, so it is not easy to tell which ensemble is state-of-the-art in this regard. This phenomenon is because different ensembles have similar overall performance but adopt different trade-offs between coverage and selective risk. In this case, we need a comprehensive metric, e.g., AURC, to identify the state-of-the-art (see Table 1).

## E  EMPIRICAL RESULTS OF SN

With the hyperparameter $c$ of 0.9, we report the selective risks given the coverage of 90% of SN ensembles and the individual SN in Figure 5. The coverage is set to 90% because the target coverage of the SN is $c$ (Geifman & El-Yaniv, 2019)), and $c$ is 0.9 in our experiments. The results show that each ensemble of SN has a lower selective risk than the individual SN.

## F  FURTHER PROPERTIES OF SELECTIVE CLASSIFIER ENSEMBLE

### F.1  THE EFFECT OF NUMBER OF MEMBERS ON SELECTIVE CLASSIFIER ENSEMBLE

We evaluate AURCs of the SR ensemble, Gambler ensemble, and SAT ensemble of different numbers of members on CIFAR10, and find that an ensemble with more members has a better performance, but is less efficient. The results are shown in Figure 6. In most cases, the AURC on the test set of CIFAR-10 decreases as the number of members in the ensemble increases. In addition, as the number of members in the ensemble grows, the effect of adding one member drops. On the one hand, the result shows that an ensemble with a small number of members has good selective classification performance. On the other hand, it indicates that when the number of member models is large, increasing the number of members to improve the performance of the selective classification ensemble is inefficient.

### F.2  GOOD CLASSIFICATION PERFORMANCE DOES NOT IMPLY GOOD SELECTIVE CLASSIFICATION PERFORMANCE

It is well known that the ensemble has better classification performance than an individual model, but this does not guarantee a better selective classification performance of the ensemble. To demonstrate this, we design an SR model with a big backbone, and show that it has as good classification

Table 3: The selective risks of ensembles under coverage 10%-100% on image classification datasets. The means and standard deviations are calculated over three trials. The best entries and those that overlap with the best entries are marked in bold.

| Dataset | coverage (%) | SR ensemble | Gambler ensemble | SAT ensemble | Reg-curr ensemble |
|---|---|---|---|---|---|
| CIFAR-10 | 100 | **5.31±0.03** | **5.29±0.03** | 5.47±0.04 | 5.74±0.07 |
| | 90 | **1.68±0.02** | 1.99±0.01 | 2.15±0.06 | 1.89±0.02 |
| | 80 | **0.45±0.05** | **0.51±0.02** | 0.63±0.03 | 0.61±0.09 |
| | 70 | **0.17±0.01** | 0.21±0.01 | 0.26±0.01 | **0.18±0.02** |
| | 60 | 0.11±0.01 | 0.18±0.03 | 0.17±0.01 | **0.08±0.00** |
| | 50 | 0.11±0.01 | 0.14±0.02 | 0.11±0.01 | **0.07±0.01** |
| | 40 | 0.12±0.03 | 0.15±0.02 | **0.06±0.01** | **0.07±0.01** |
| | 30 | 0.13±0.05 | 0.13±0.03 | **0.06±0.02** | **0.08±0.03** |
| | 20 | 0.12±0.02 | 0.17±0.05 | **0.00±0.00** | 0.02±0.02 |
| | 10 | 0.10±0.08 | 0.14±0.05 | **0.00±0.00** | **0.00±0.00** |
| SVHN | 100 | 2.44±0.01 | 2.42±0.02 | **2.36±0.01** | 2.41±0.01 |
| | 90 | 0.59±0.00 | 0.60±0.03 | **0.50±0.01** | 0.54±0.02 |
| | 80 | 0.42±0.03 | 0.38±0.01 | **0.34±0.01** | 0.39±0.02 |
| | 70 | 0.34±0.02 | **0.32±0.01** | **0.31±0.01** | 0.35±0.01 |
| | 60 | 0.32±0.02 | 0.30±0.01 | **0.28±0.01** | 0.35±0.01 |
| | 50 | 0.29±0.02 | **0.26±0.01** | **0.26±0.00** | 0.30±0.01 |
| | 40 | **0.25±0.02** | **0.27±0.02** | **0.25±0.01** | 0.28±0.01 |
| | 30 | **0.22±0.03** | 0.26±0.01 | **0.20±0.01** | 0.25±0.03 |
| | 20 | 0.22±0.01 | 0.26±0.01 | **0.18±0.02** | **0.19±0.03** |
| | 10 | 0.21±0.02 | 0.23±0.00 | **0.17±0.02** | **0.18±0.03** |
| CIFAR-100 | 100 | **24.66±0.08** | 25.50±0.05 | 25.23±0.13 | 25.70±0.09 |
| | 90 | **19.15±0.15** | 19.88±0.05 | 19.77±0.28 | 20.16±0.14 |
| | 80 | **14.32±0.22** | 15.75±0.09 | 15.00±0.20 | 15.22±0.07 |
| | 70 | **9.78±0.13** | 12.11±0.18 | 10.29±0.24 | 10.41±0.38 |
| | 60 | **5.81±0.06** | 8.89±0.16 | 6.43±0.20 | 6.58±0.27 |
| | 50 | **2.95±0.04** | 6.22±0.10 | 3.41±0.15 | 3.45±0.05 |
| | 40 | **1.40±0.13** | 4.37±0.06 | 1.96±0.13 | 1.74±0.11 |
| | 30 | **0.75±0.05** | 2.67±0.01 | 1.13±0.02 | 0.89±0.06 |
| | 20 | **0.62±0.06** | 1.91±0.04 | **0.72±0.06** | **0.62±0.04** |
| | 10 | 0.33±0.09 | 1.42±0.16 | 0.57±0.09 | **0.13±0.05** |

performance as an SR ensemble with a standard backbone but worse selective classification performance than an SR model with a standard backbone. The big backbone is designed to have twice as many filters in every convolutional layer and neurons in every fully connected hidden layer as those of the standard VGG-16, which is therefore called *Big VGG-16*. It is easy to see that its number of parameters is approximately $2^2 = 4$ times as many as that of standard VGG-16. We train an SR ensemble of 4 VGG-16s and an SR model with a backbone of Big VGG-16 on CIFAR-10 and show the evaluation results in Figure 7 and Table 5. Figure 7 shows that when coverage is high, the ensemble and the big individual model have similar selective risks, and especially, the classification error rates (i.e., selective risk of 100% coverage) of the ensemble and the big individual model are similar. However, when coverage is low, the big individual model has significantly higher selective risk than the ensemble. Table 5 shows that the AURC of Big VGG-16 is much higher than the ensemble of 4 VGG-16s and even higher than SR. In summary, we show that a selective classifier with a good classification performance is not guaranteed to have good selective classification performance, so the good selective classification performance of the ensemble is not a trivial result of its good classification performance.

### F.3 THE EFFECTS OF LABEL NOISE OF SVHN ON SELECTIVE CLASSIFIER ENSEMBLES

In this section, we compare the effect of label noise of SVHN on the SR ensemble with that on SAT ensemble, whose result might explain the abnormal experimental results (compared to results on other datasets) on SVHN in Section 6.2. SVHN is not a clean dataset, and much more label

Table 4: The selective risks of ensembles under coverage 10%-100% on text classification datasets. The means and standard deviations are calculated over three trials. The best entries and those that overlap with the best entries are marked in bold.

| Dataset | coverage (%) | SR ensemble | Gambler ensemble | SAT ensemble | Reg-curr ensemble |
|---|---|---|---|---|---|
| | 100 | 14.13±0.23 | 14.62±0.23 | **13.64±0.23** | 15.28±0.31 |
| | 90 | **11.41±1.02** | 11.50±0.46 | **10.69±0.13** | 11.68±0.44 |
| | 80 | **7.75±0.29** | 9.99±0.88 | **8.46±0.63** | 8.36±0.29 |
| | 70 | **6.41±0.33** | 8.51±0.72 | 7.93±0.44 | **6.64±0.29** |
| | 60 | **5.71±0.33** | 7.35±1.20 | 6.39±0.19 | **6.12±0.33** |
| MRPC | 50 | **4.08±1.29** | **4.41±0.40** | **3.59±1.01** | **4.74±0.23** |
| | 40 | **3.25±0.29** | **3.86±0.76** | **3.25±0.76** | **3.66±0.50** |
| | 30 | **3.25±0.00** | **3.52±0.38** | **3.52±0.38** | **2.98±0.77** |
| | 20 | **2.44±1.72** | **3.25±0.58** | **3.25±0.58** | **3.66±0.00** |
| | 10 | **3.25±2.30** | 4.88±0.00 | **1.63±1.15** | 5.69±1.15 |
| | 100 | 8.16±0.04 | **8.18±0.20** | **8.03±0.09** | 8.17±0.01 |
| | 90 | **4.74±0.04** | 5.04±0.14 | 5.03±0.09 | **4.74±0.12** |
| | 80 | **2.94±0.11** | 3.08±0.08 | 2.97±0.04 | **2.98±0.01** |
| | 70 | **1.84±0.10** | 1.91±0.04 | 1.92±0.11 | **1.88±0.06** |
| | 60 | **1.20±0.05** | 1.27±0.05 | 1.36±0.01 | 1.30±0.08 |
| QNLI | 50 | **1.04±0.05** | **1.04±0.08** | 1.13±0.03 | **0.98±0.03** |
| | 40 | **0.72±0.02** | **0.73±0.04** | 1.02±0.11 | **0.70±0.06** |
| | 30 | **0.45±0.03** | **0.51±0.03** | 0.83±0.08 | **0.43±0.09** |
| | 20 | **0.30±0.11** | **0.37±0.15** | 0.67±0.04 | **0.30±0.09** |
| | 10 | 0.30±0.09 | **0.12±0.09** | **0.61±0.34** | **0.12±0.09** |
| | 100 | **15.04±0.06** | **14.82±0.15** | 15.03±0.06 | **14.89±0.14** |
| | 90 | **11.01±0.11** | 11.78±0.09 | 11.37±0.06 | **11.21±0.17** |
| | 80 | **7.93±0.08** | 9.62±0.08 | 8.26±0.20 | **8.02±0.07** |
| | 70 | **5.81±0.04** | 8.03±0.23 | **5.81±0.12** | **5.85±0.22** |
| | 60 | **4.28±0.07** | 6.41±0.25 | **4.08±0.19** | **4.05±0.10** |
| MNLI | 50 | **3.22±0.04** | 4.95±0.10 | **2.99±0.18** | **3.04±0.10** |
| | 40 | **2.69±0.11** | 3.57±0.21 | **2.08±0.03** | 2.22±0.06 |
| | 30 | **2.13±0.14** | 2.35±0.18 | **1.57±0.03** | 1.75±0.06 |
| | 20 | **1.34±0.10** | **1.53±0.17** | **1.39±0.13** | **1.36±0.06** |
| | 10 | 0.98±0.05 | 1.32±0.14 | **0.71±0.22** | **0.71±0.08** |

Table 5: The AURCs($/10^{-4}$) of Big VGG-16, a vanilla VGG-16, and the ensemble of 4 VGG-16s on CIFAR-10. The best entries are marked in bold.

| Dataset | Big VGG-16 | VGG-16 | Ensemble |
|---|---|---|---|
| CIFAR-10 | 89.2 | 69.6 | **49.3** |

noise can be detected in SVHN than in CIFAR-10 and CIFAR-100. Using the soft label of SAT (Huang et al., 2020), we detect label noise in SVHN, CIFAR-10, and CIFAR-100, and find that SVHN has significantly more label noise than CIFAR-10 and CIFAR-100. The result is presented in the following. In addition, it is known that SAT is robust to label noise (Huang et al., 2020), while SR is not so, so we conjecture that the label noise of SVHN is why the SR ensemble is inferior to SAT on SVHN.

We detect label noise with the help of the soft label of SAT. For a sample $x_i$, the soft label of SAT (Huang et al., 2020), $t_{i,y_i}$, is used to measure $x_i$'s learning difficulty. The soft label of SAT is initialized as 1 and updated at every training epoch as below

$$t_{i,y_i} \leftarrow \alpha \times t_{i,y_i} + (1 - \alpha) \times p_\theta(y_i|x_i),$$

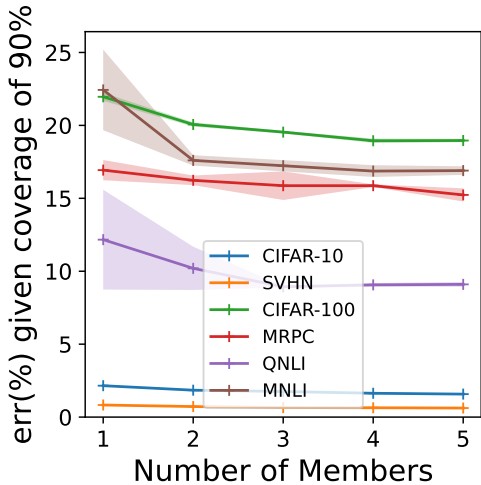

Figure 5: Selective risks of SN ensembles (with 2 to 5 members) and the individual SN (with only 1 member) given the coverage of 90%

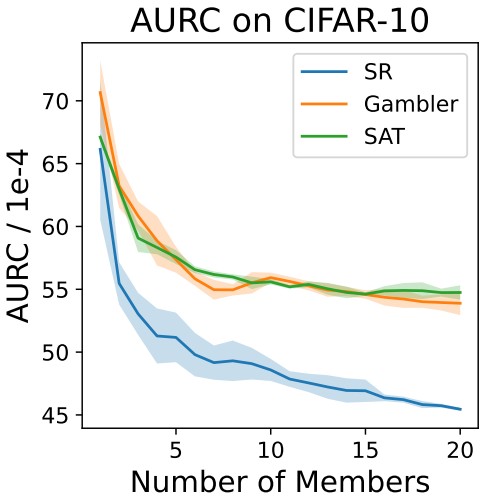

Figure 6: The AURCs on the test set of CIFAR-10 of the SR ensemble, Gambler ensemble, and SAT ensemble of different numbers of members

where $p_\theta(Y|x)$ is the predictive probability distribution of the classifier, $y_i$ is the label of $x_i$, $\alpha$ is a hyperparameter. The smaller the $t_{i,y_i}$ is, the lower the true class predictive probability of the classifier on $x_i$ during training time, indicating that $x_i$ is more difficult to learn. By selecting a percentage of samples with the lowest $t_{i,y_i}$, we get the most difficult samples to learn for the classifier, from which we can easily detect label noise manually.

In training sets of SVHN, CIFAR-10, and CIFAR-100, we detect label noise manually among the top-0.1% difficult (measured by the soft label of SAT) samples. The numbers of mislabeled samples detected in SVHN, CIFAR-10, and CIFAR100 are shown in Table 6. The result shows that SVHN has significantly more mislabeled samples detected than CIFAR-10 and CIFAR-100, indicating much more label noise in SVHN than in CIFAR-10 and CIFAR-100.

To verify the effect of label noise, the following experiments are designed. Firstly, we detect label noise manually among the 1% of the hardest-to-learn samples of SVHN training set and test set,

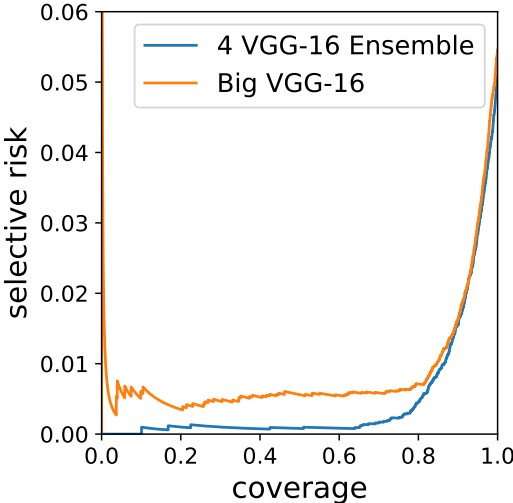

Figure 7: Risk-coverage curves of the ensemble of 4 VGG-16s and the Big VGG-16 on CIFAR-10

Table 6: Numbers of mislabeled samples in the top-0.1% difficult training samples of SVHN, CIFAR-10, and CIFAR-100.

| Dataset | #Mislabeled | #Top-0.1% | Proportion |
|---------|-------------|-----------|------------|
| SVHN | 73 | 73 | 100% |
| CIFAR-10 | 1 | 50 | 2% |
| CIFAR-100 | 1 | 50 | 2% |

using the soft label of SAT. Secondly, we remove the detected mislabeled samples from the original dataset. The remaining SVHN dataset is called the *clean SVHN*. Accordingly, the original dataset is called the *original SVHN*. Finally, we retrain and test the SR ensemble and SAT ensemble and compare their test results. In the second step, the reason for removing mislabeled samples rather than modifying them is that some samples cannot be classified even by humans, and some samples are not in the range of categories of SVHN. Thus, the label noise cannot be eliminated by modifying the labels but by removing mislabeled samples.

The test results of the SR ensemble and SAT ensemble on clean SVHN are shown in Table 7. It is not surprising that the AURCs of the SR ensemble and SAT ensemble are significantly lower on the clean SVHN than the original SVHN. Furthermore, on the clean SVHN, when the number of members is 5, the AURC of the SR ensemble is lower than that of SAT ensemble. Combined with results on the original SVHN, where the AURC of the SR ensemble is higher than that of SAT ensemble, we conclude that label noise in SVHN is why the SR ensemble has a higher AURC than SAT ensemble. In other words, label noise is why the SR ensemble performs worse in selective classification than SAT ensemble on SVHN.

Table 7: AURC/$10^{-4}$ of SR ensemble and SAT ensemble on the clean SVHN

| Dataset | #Member | SR | SAT |
|---------|---------|------|------|
| | 1 | 12.3 | **7.8** |
| | 2 | 8.2 | **7.1** |
| clean SVHN | 3 | 7.3 | **6.8** |
| | 4 | **6.8** | **6.8** |
| | 5 | **6.4** | 6.8 |

In summary, by experiments, we show that the SR ensemble is not as robust to label noise as SAT ensemble, and label noise in SVHN is why the SR ensemble is not as good as SAT ensemble on SVHN. We construct the *clean SVHN*, which is SVHN without some mislabeled samples. On the clean SVHN, we compare the SR ensemble with SAT ensemble and find that the SR ensemble is superior to SAT ensemble in selective classification performance. Combined with former experimental results, we conclude that label noise in SVHN is why the SR ensemble is inferior to SAT on SVHN.

Considering the experimental results on the clean SVHN and previous experimental results on CIFAR-10 and CIFAR-100 (see Table 7 and Table 1), the SR ensemble is superior to SAT ensemble in selective classification on clean image classification datasets, Thus, SR ensemble is the state-of-the-art selective classification method on clean image classification datasets, but is not as robust to label noise as SAT ensemble.

## G  AN EXTENSION OF THEOREM 2

This section discusses the lower bound of $\phi_0$ mentioned in Theorem 2. We aim to calculate $\phi_0$'s lower bound without training an ensemble (otherwise, we can measure it directly on the ensemble).

**Preliminaries.** The $\phi_0$ can be obtained by solving the following optimization problem.

$$\max_{\phi} \phi$$
$$\text{s.t. } R_{\mathrm{ens}}(\phi) < R(\phi),$$

where $R_{\mathrm{ens}}(\phi)$ and $R(\phi)$ are the selective risks of the ensemble and the individual model under coverage $\phi$, respectively. Suppose *we know all about the individual model*, e.g., the mapping from $\phi$ to $R$ is known. Since maximizing coverage is equivalent to minimizing the confidence threshold for a fixed model, the original optimization problem can be substituted by

$$\min_{\tau, \tau_{\mathrm{ens}}} \tau \tag{42}$$
$$\text{s.t. } \phi_{\mathrm{ens}}(\tau_{\mathrm{ens}}) = \phi(\tau)$$
$$R_{\mathrm{ens}}(\phi_{\mathrm{ens}}(\tau_{\mathrm{ens}})) < R(\phi(\tau)),$$

where $\phi_{\mathrm{ens}}(\tau_{\mathrm{ens}})$ is coverage of the ensemble with confidence threshold $\tau_{\mathrm{ens}}$, and $\phi(\tau)$ is coverage of the individual model given the confidence threshold $\tau$, and then $\phi_0 = \phi(\tau^*)$, where $\tau^*$ is the optimal solution to (42). Assume that $R$ *is a monotone increasing function* and that $\phi$ *is a monotone decreasing function*[5], then it is easy to show, using proof by contradiction, that (42) can be further transformed into

$$\min_{\tau, \tau_{\mathrm{ens}}} \tau \tag{43}$$
$$\text{s.t. } \phi_{\mathrm{ens}}(\tau_{\mathrm{ens}}) \geq \phi(\tau)$$
$$R_{\mathrm{ens}}(\phi_{\mathrm{ens}}(\tau_{\mathrm{ens}})) < R(\phi(\tau)).$$

To solve (43), we need more information about the ensemble. Besides the number of classes $K$, assume that we know: *an oracle that tells whether a sample is definite; $M$, the number of member models; and $B$, the upper bound of $p_{\Pi_1^k,...,\Pi_M^k}(\cdot|A)$ for all $k \in \{1, 2, ..., K\}$*. It is natural to know $K$ and $M$, and we need the oracle and $B$ because they provide critical information about the ensemble's behavior. The oracle can be implemented by an ensemble with $M'$ ($M' \ll M$) members.

**Eliminating the Unknowns.** We are now committed to translating the unknowns in the (43) into known quantities. Firstly, we eliminate the unknowns in the first constraint. According to (20) and Lemma 4, it is easy to prove (by an integral)

$$\Pr(\mathrm{C}_{\mathrm{ens}} \geq \tau_{\mathrm{ens}}|A) \leq \beta(1 - \tau_{\mathrm{ens}})^M, \tag{44}$$

where C and $\mathrm{C}_{\mathrm{ens}}$ are the confidence scores of the individual model and the ensemble, respectively, $A/D$ represents the event that the input sample is ambiguous/definite, $\beta = K \cdot M^{M-1} \cdot B$. Combining

---

[5]The latter is actually an obvious fact.

(44) with $\phi_{\text{ens}} = \Pr(C \geq \tau_{\text{ens}}, A) + \Pr(C \geq \tau_{\text{ens}}, D) = \Pr(C \geq \tau_{\text{ens}}|A)\Pr(A) + \Pr(C \geq \tau_{\text{ens}}, D)$, we have

$$\Pr(C \geq \tau_{\text{ens}}, D) \leq \phi_{\text{ens}}(\tau_{\text{ens}}) \leq \beta(1 - \tau_{\text{ens}})^M \Pr(A) + \Pr(C \geq \tau_{\text{ens}}, D). \tag{45}$$

Thus, we can intensify the first constraint as

$$\phi(\tau) \leq \Pr(C \geq \tau_{\text{ens}}, D). \tag{46}$$

Secondly, we eliminate the unknowns in the second constraint. according to (41), (35) is a sufficient condition of $R_{\text{ens}} < R$. We rewrite (35) as

$$\Pr(A|C_{\text{ens}} \geq \tau_{\text{ens}}) + \Pr(Err|D, C \geq \tau_{\text{ens}}) < R(\phi(\tau)), \tag{47}$$

where $C_{\text{ens}}$ is the confidence score of the ensemble, and $Err$ represents the event that the individual model makes an error prediction. Note that the first term in the second constraint of (47) contains $C_{\text{ens}}$, which is unknown, so we cannot directly replace (43)'s second constraint with (47). We eliminate $C_{\text{ens}}$ as follows:

$$
\begin{aligned}
\Pr(A|C_{\text{ens}} \geq \tau_{\text{ens}}) &= \frac{\Pr(A, C_{\text{ens}} \geq \tau_{\text{ens}})}{\Pr(C_{\text{ens}} \geq \tau_{\text{ens}})} \\
&= \frac{\Pr(A, C_{\text{ens}} \geq \tau_{\text{ens}})}{\phi_{\text{ens}}(\tau_{\text{ens}})} \\
&= \frac{\Pr(A) \cdot \Pr(C_{\text{ens}} \geq \tau_{\text{ens}}|A)}{\phi_{\text{ens}}(\tau_{\text{ens}})} \\
&\leq \frac{\Pr(A) \cdot \beta(1 - \tau_{\text{ens}})^M}{\phi_{\text{ens}}(\tau_{\text{ens}})} \\
&\leq \frac{\Pr(A) \cdot \beta(1 - \tau_{\text{ens}})^M}{\phi(\tau)},
\end{aligned}
$$

where the first inequality is due to (44), and the second inequality is due to (45) and (46). Thus, a sufficient condition of (47) is

$$\frac{\Pr(A) \cdot \beta(1 - \tau_{\text{ens}})^M}{\phi(\tau)} + \Pr(Err|D, C \geq \tau_{\text{ens}}) < R(\phi(\tau)), \tag{48}$$

with which we replace the second constraint of (43). In summary, we can intensify the constraints of (43) and obtain the following optimization problem that does not contain the unknowns. It is easy to see that the optimal solution to (49) is an upper bound of that to (43).

$$\min_{\tau, \tau_{\text{ens}}} \tau \tag{49}$$
$$\text{s.t. } \phi(\tau) \leq \Pr(C \geq \tau_{\text{ens}}, D)$$
$$\frac{\Pr(A) \cdot \beta(1 - \tau_{\text{ens}})^M}{\phi(\tau)} + \Pr(Err|D, C \geq \tau_{\text{ens}}) < R(\phi(\tau)).$$

**Further Simplification and Final Result.** It is easy to show, by proof of contraction, that the first constraint of (49) can be substituted by

$$\phi(\tau) = \Pr(C \geq \tau_{\text{ens}}, D).$$

Thus, the second constraint of (49) can be simplified as

$$
\begin{aligned}
&\Pr(A) \cdot \beta(1 - \tau_{\text{ens}})^M + \Pr(Err|D, C \geq \tau_{\text{ens}})\phi(\tau) < R(\phi(\tau))\phi(\tau) \\
\Leftrightarrow &\Pr(A) \cdot \beta(1 - \tau_{\text{ens}})^M + \Pr(Err|D, C \geq \tau_{\text{ens}})\Pr(D, C \geq \tau_{\text{ens}}) < R(\phi(\tau))\phi(\tau) \\
\Leftrightarrow &\Pr(A) \cdot \beta(1 - \tau_{\text{ens}})^M + \Pr(Err, D, C \geq \tau_{\text{ens}}) < R(\phi(\tau))\phi(\tau) \\
\Leftrightarrow &\Pr(A) \cdot \beta(1 - \tau_{\text{ens}})^M + \Pr(Err, D, C \geq \tau_{\text{ens}}) < \Pr(Err, C \geq \tau).
\end{aligned}
$$

Thus, the final version of the optimization problem with respect to $\tau$ is

$$\min_{\tau, \tau_{\text{ens}}} \tau \tag{50}$$
$$\text{s.t. } \phi(\tau) = \Pr(C \geq \tau_{\text{ens}}, D)$$
$$\Pr(A) \cdot \beta(1 - \tau_{\text{ens}})^M + \Pr(Err, D, C \geq \tau_{\text{ens}}) < \Pr(Err, C \geq \tau).$$

---

**Algorithm 1:** A Lower Bound of $\phi_0$.

---

**Input:** the individual model $\theta$; the test set $\mathcal{D} = \{(x_i, y_i)\}_{i=1}^N$; the number of classes $K$; the oracle $\Omega : \mathcal{X} \to \{0, 1\}$ that tells whether a sample is definite; the number of member models $M$; and $B$, the upper bound of $p_{\Pi_1^k, \ldots, \Pi_M^k}(\cdot | A)$ for all $k \in \{1, 2, \ldots, K\}$.

**Output:** An lower bound of $\phi_0$ mentioned in Theorem 2

$left = 0$
$right = 1$
$\epsilon = 10^{-9}$
**while** $right - left > \epsilon$ **do**
    $\tau = (left + right)/2$
    $\tau_{\text{ens}} = \text{SEARCHFORTAUENS}(\tau, \theta, \mathcal{D}, \Omega)$
    **if** $\tau_{\text{ens}}$ is not None and $\text{VERIFYSECONDCONSTRAINT}(\tau, \tau_{\text{ens}}, \theta, \mathcal{D}, \Omega, K, M, B)$ is True
    **then**
        |  $right = \tau$
    **else**
        |  $left = \tau$
$\tau^* = (left + right)/2$
**return** $\frac{1}{N} \sum_{i=1}^N \mathbb{I}\{C(x_i; \theta) \geq \tau^*\}$ // $C(x_i; \theta)$ is the confidence of $\theta$ on sample $x_i$.

---

Suppose $\phi_0 = \phi(\tau_0)$, since $\tau_0$ is the optimal solution to (43), the optimal solution to (50) (denoted as $\tau^*$) provides an upper bound of $\tau_0$. Thus, considering $\phi$ is a monotone decreasing function of $\tau$, $\phi(\tau^*)$ is a lower bound of $\phi_0 = \phi(\tau_0)$.

**Algorithm.**

We design Algorithm 1 to search for the solution to (50) and then obtain the lower bound of $\phi_0$. Since $\tau_{\text{ens}}$ is determined by $\tau$ (the first constraint of (50)), (50) can be reduced to a one-dimensional search problem. Our algorithm adopts a binary search for efficiency, although this method might provide a suboptimal solution. The procedure of Algorithm 1 in each iteration of the binary search is as follows.

1. Given current $\tau$, Algorithm 1 determines $\tau_{\text{ens}}$ using SEARCHFORTAUENS (see Algorithm 2), a procedure that searches for $\tau_{\text{ens}} \in [0, 1]$ using binary search s.t. $\phi(\tau) = \Pr(C \geq \tau_{\text{ens}}, D)$. Note that $\tau_{\text{ens}}$ might not exist, as long as $\tau$ is so low that $\phi(\tau) > \Pr(D) = \sup_{\tau_{\text{ens}}} \Pr(C \geq \tau_{\text{ens}}, D)$. This problem will be addressed shortly.

2. Algorithm 1 exams whether $\tau_{\text{ens}}$ exists. If $\tau_{\text{ens}}$ exists, Algorithm 1 then examines whether the second constraint of (50) holds for current $\tau$ and $\tau_{\text{ens}}$, which is implemented by VERIFYSECONDCONSTRAINT (see Algorithm 3).

3. If $\tau_{\text{ens}}$ exists and the second constraint holds, Algorithm 1 searches for a smaller $\tau$ in the left half feasible area; otherwise, Algorithm 1 searches for a greater $\tau$ in the right half feasible area.

Once the binary search completes and outputs $\tau^*$, Algorithm 1 returns the coverage of $\theta$ with confidence threshold $\tau^*$.

**An Example.**

To show that Algorithm 1 works in reality, we run this algorithm on CIFAR-10, using the same individual model as Section 6. In this example, $K = 10$, $M = 5$, the oracle is implemented by another ensemble with two individual models (the oracle outputs True if and only if the STD over member models' predictive distributions $< 10^{-3}$). Note that it is difficult to estimate $B$. On the one hand, we need to train an ensemble with $M$ models to estimate $B$, which is costly. On the other hand, the domain of $p_{\Pi_1^k, \ldots, \Pi_M^k}(\cdot | A)$ has high dimension, so the observed data points are sparse in this domain, which makes the estimation of $B$ more difficult. Thus, we do not estimate $B$ but try several hypothetical values of $B$ to see at what $B$ the lower bound of $\phi_0$ is big. With different $B$s, we obtain different lower bounds of $\phi_0$ as Table 8 shows. We can see that when $B \leq 10^8$, the lower bound of $\phi_0$ is greater than 50%, which indicates that Algorithm 1 may be robust to the choice of $B$.

---

**Algorithm 2:** SEARCHFORTAUENS

---

**Input:** the confidence threshold $\tau$; the individual model $\theta$; the test set $\mathcal{D} = \{(x_i, y_i)\}_{i=1}^N$; the oracle $\Omega : \mathcal{X} \to \{0, 1\}$ that tells whether a sample is definite.
**Output:** $\tau_{\text{ens}} \in [0, 1]$ that satisfies the first constraint of (50).

$\phi = \frac{1}{N} \sum_{i=1}^N \mathbb{I}\{C(x_i; \theta) \geq \tau\}$ // $C(x_i; \theta)$ is the confidence of $\theta$ on sample $x_i$.
**if** $\phi > \frac{1}{N} \sum_{i=1}^N \Omega(x_i)$ **then**
    ⌊ **return** *None*
$left = 0$
$right = 1$
$\epsilon = 10^{-9}$
**while** $right - left > \epsilon$ **do**
    |  $\tau_{\text{ens}} = (left + right)/2$
    | **if** $\frac{1}{N} \sum_{i=1}^N \mathbb{I}\{C(x_i; \theta) \geq \tau_{\text{ens}}\} \cdot \Omega(x_i) < \phi$ **then**
    |  | $right = \tau_{\text{ens}}$
    | **else**
    |  ⌊ $left = \tau_{\text{ens}}$
**return** $(left + right)/2$

---

**Algorithm 3:** VERIFYSECONDCONSTRAINT

---

**Input:** the confidence threshold $\tau$; $\tau_{\text{ens}}$; the individual model $\theta$; the test set $\mathcal{D} = \{(x_i, y_i)\}_{i=1}^N$; the oracle $\Omega : \mathcal{X} \to \{0, 1\}$ that tells whether a sample is definite; the number of classes $K$; the number of member models $M$; and $B$, the upper bound of $p_{\Pi_1^k, \dots, \Pi_M^k}(\cdot | A)$ for all $k \in \{1, 2, \dots, K\}$.
**Output:** True if and only if $\tau$ and $\tau_{\text{ens}}$ satisfy the second constraint of (50)

$P_A = \frac{1}{N} \sum_{i=1}^N [1 - \Omega(x_i)]$
$\beta = K \cdot M^{M-1} \cdot B$
$leftHandSide = P_A \cdot \beta (1 - \tau_{\text{ens}})^M + \frac{1}{N} \sum_{i=1}^N \mathbb{I}\{f(x_i; \theta) \neq y_i\} \cdot \Omega(x_i) \cdot \mathbb{I}\{C(x_i; \theta) \geq \tau_{\text{ens}}\}$
$rightHandSide = \frac{1}{N} \sum_{i=1}^N \mathbb{I}\{f(x_i; \theta) \neq y_i\} \cdot \mathbb{I}\{C(x_i; \theta) \geq \tau\}$
**return** $\mathbb{I}\{leftHandSide < rightHandSide\}$

---

This example also indicates the relationship between the ensemble's diversity and its selective classification performance. Since an ensemble with a smaller $B$ seems to have more diversity over ambiguous samples, the result in Table 8 suggests that as long as the ensemble has enough diversity over ambiguous samples, the ensemble is guaranteed to have a lower selective risk than the individual model under a considerable range of coverage.

Table 8: The relationship between $\phi_0$'s lower bound and $B$ on CIFAR-10, where the individual model is the same as Section 6.

| B | 1 | 10 | $10^2$ | $10^3$ | $10^4$ | $10^5$ | $10^6$ | $10^7$ | $10^8$ | $10^9$ | $10^{10}$ | $10^{11}$ | $10^{12}$ | $10^{13}$ | $10^{14}$ |
|---|---|---|---|---|---|---|---|---|---|---|---|---|---|---|---|
| lower bound of $\phi_0$ | 0.737 | 0.737 | 0.737 | 0.737 | 0.736 | 0.735 | 0.729 | 0.705 | 0.599 | 0 | 0 | 0 | 0 | 0 | 0 |

