# OpenReview forum: "Selective Classifier Ensemble"
_ICLR.cc/2023/Conference — Submitted to ICLR 2023_

### Official Review · Reviewer_KH7o · 2022-10-21

**Confidence:** 2
**Correctness:** 3
**Technical Novelty And Significance:** 3
**Empirical Novelty And Significance:** 2
**Recommendation:** 6

**Clarity, Quality, Novelty And Reproducibility:**

The submission is well written and great effort has been taken to spell out all steps of the arguments. The main finding of the submission, that ensemble learning enables more accurate selective classification than using a single model, is unsurprising, but the theory presented in the submission appears to be novel. It is also interesting to see that selective classification based on maximum probability, the most obvious approach, outperforms the state-of-the-art when care is taken to use the same experimental setup for both approaches.

**Strength And Weaknesses:**

+ selective classification is an interesting and important problem
+ results challenge what is considered the state of the art
+ convincing empirical findings with theoretical support

- the proposed method is not new and the improvements observed with ensemble learning are unsurprising
- no discussion of work on estimating epistemic uncertainty, which seems a highly relevant topic

Questions and comments:

Are the numbers after "+/-" in Table 1 standard deviations? How were these obtained?

"there has been no systematic study of ensemble methods in selective classification" - repeated verbatim in two adjacent sentences

"selective risk ... are"

"which detects samples different greatly from a given dataset" - rephrase

"to the overconfident problem" - rephrase

"For definite samples, given the predictive probability distribution of one member model..." - I could not follow the reasoning here

"AURCs of the individual model"

"for the better performance of the ensemble than the individual selective classifier" - rephrase



**Summary Of The Paper:**

The submission shows that simple randomization-based learning of deep ensembles improves on training a single model when performing selective classification, regardless of the selective classification method that is employed. Given some assumptions that seem to hold in most of the classification scenarios considered in the submission, it is proven that this must indeed be the case. Another, empirical, finding of the submission is that simple selective classification based on a threshold on the maximum class probability is preferable to other methods, including the current state-of-the-art.

**Summary Of The Review:**

Although the main finding of the submission, that randomization-based ensembles outperform individual models, is unsurprising, particularly considering the experiments reported by Lakshminarayanan et al., it may be useful to have the additional theoretical justification for this method that the submission presents. This justification, based on some assumptions that seem reasonable (and seem to hold in pertinent benchmark problems), appears to be novel. For practitioners, it is also useful to see that selective classification based on maximum probability beats the state of the art in the experiments presented in the submission.

---

> ### Author Response · Authors · 2022-11-18
> **Response to Reviewer KH7o**
>
> Dear reviewer, thanks for your careful and valuable comments. The contents of this response include:
>
> 1. clarification on the relationship between our work and epistemic uncertainty;
> 2. clarification on the notation "+/-" in Table 1;
> 3. response to grammar mistakes;
> 4. clarification on Assumption 3.
>
> Following are our answers to each concern.
>
> > no discussion of work on estimating epistemic uncertainty, which seems a highly relevant topic
>
> This is relevant if we measure the uncertainty by the entropy of the model's predictive probability distribution. In this case, the epistemic uncertainty is the difference between the uncertainty of the ensemble and that of the individual model, i.e., the uncertainty of the ensemble = epistemic uncertainty + expected uncertainty of the individual model [1].
>
> However, we measure the uncertainty not by entropy but by the maximum output of the softmax layer. In this case, the equation above does not hold, and epistemic uncertainty does not help us understand why the ensemble method works in selective classification.
>
> > Are the numbers after "+/-" in Table 1 standard deviations? How were these obtained?
>
> Yes. They are estimated over 3 trials. We will clarify these in the revision (see the updated caption of Table 1).
>
> > "there has been no systematic study of ensemble methods in selective classification" - repeated verbatim in two adjacent sentences
> > "selective risk ... are"
> > "which detects samples different greatly from a given dataset" - rephrase
> > "to the overconfident problem" - rephrase
> > "AURCs of the individual model"
> > "for the better performance of the ensemble than the individual selective classifier" - rephrase
>
> Thanks for your careful review. These grammar mistakes and diction problems will be fixed in the revision.
>
> > "For definite samples, given the predictive probability distribution of one member model..." - I could not follow the reasoning here
>
> This is due to the definition of definite samples. The definite samples are those for which all member models provide the same predictive probability distribution. Hence, given a definite sample and a member model's predictive probability distribution $\boldsymbol{\pi}_0$, other member models' predictive probability distributions must be $\boldsymbol{\pi}_0$. This is an idealization of the real situation to reflect that models have similar predictions on definite samples.
>
> [1] Andrey Malinin, Bruno Mlodozeniec, and Mark Gales. Ensemble distribution distillation. In ICLR, 2020.

---

> > ### Comment · Reviewer_KH7o · 2022-11-24
> > **Acknowledgement of response to Reviewer KH7o**
> >
> > Thank you for your response. Regarding epistemic uncertainty, I appreciate your reply, and it seems that it would be worthwhile to state this in the paper. This reply brings up the question of why entropy is not considered for selective classification in the paper, which should also be addressed.

---

> > > ### Author Response · Authors · 2022-11-26
> > > **Response to Reviewer KH7o's reply**
> > >
> > > Dear reviewer, thanks for your valuable comments. We will add the discussion about epistemic uncertainty in the revision. This paper does not consider entropy because:
> > > 1. the entropy-based method, which takes the predictive distribution’s entropy as the confidence score, does not have an outstanding selective classification performance;
> > > 2. this paper focuses on why the ensemble works rather than exploring ensembles of all kinds of selective classifiers.
> > >
> > > First, the entropy-based method is not an outstanding selective classification method. Liu et al. [1] show that the entropy-based method is not as effective as Deep Gambler (one of the baselines in this paper). Second, this paper focuses on the ensemble of selective classifiers and the reason why the ensemble works well, so we do not explore the individual model too much. Thus, this paper does not consider the entropy-based method. We will add the discussion about works related to entropy in the revision.
> > >
> > > [1] Ziyin Liu, Zhikang Wang, Paul Pu Liang, Russ R Salakhutdinov, Louis-Philippe Morency, and Masahito Ueda. Deep gamblers: Learning to abstain with portfolio theory. In NeurIPS, 2019.

---

### Official Review · Reviewer_zrP4 · 2022-10-23

**Confidence:** 4
**Correctness:** 4
**Technical Novelty And Significance:** 1
**Empirical Novelty And Significance:** 2
**Recommendation:** 3

**Clarity, Quality, Novelty And Reproducibility:**

The paper is clear, readable, the images and plots are of good quality. However, there is no technical novelty, the contributions are only marginally significant.

**Strength And Weaknesses:**

**Strength**

- Classification with abstention has gained a lot of attention in recent years. It is an important topic.
- Interesting comparison between methods.



**Weaknesses**

- Novelty.

- Table 1: it is not clear whether the results are presented with the same level of coverage. If the coverage level is different, it is difficult to compare the values. (It is recommended to add the coverage in the table, or to specify and use a fixed coverage).

Ensemble techniques are presently hard to realize in the context of DNNs, for which it could be very costly to train sufficiently many ensemble members [1] [2].

- What is a sufficient number of ensemble members?

Figure 6 shows that the AURC  for the SR method on the test set decreases as the number of members in the ensemble increases. What is the accuracy of the model and coverage at M=20 for SR?

- The big backbone is designed to have twice as many filters in every convolutional layer (VGG-16) why the authors didn't consider a ResNet-50/101?



[1] Kush R Varshney. A risk bound for ensemble classification with a reject option. In Statistical Signal Processing Workshop (SSP), 2011 IEEE, pages 769–772. IEEE, 2011

[2] Yoav Freund, Yishay Mansour, and Robert E Schapire. Generalization bounds for averaged classifiers. Annals of Statistics, pages 1698–1722, 2004.

**Summary Of The Paper:**

In this paper the authors address the selective classification problem with the goal of reducing the selective risk (a non-convex function of the model prediction). A selective classifier is a pair (f, g), where f is a classifier, and g: X -> {0, 1} is a selection function, that allows to abstain {= 0} on some difficult input. In this paper the approach combines several weak selective classifiers (ensemble wit M = 2 to 5 members).
The contribution is to prove that an ensemble has a lower selective risk than the individual model under a range of coverage.
The method has been tested on image classification (CIFAR-10/100 and SVHN) and text classification (MRPC, MNLI, QNLI) tasks. The backbones used are a VGG-16 and BERT-base networks respectively. The method is compared against selective classification approaches.


**Summary Of The Review:**

The paper presents a series of experiments that support the hypothesis that a set of selective classifiers reduces selective risk. At the same time it suggests that the number of members of the ensemble should be low (2-5). Obviously, a high number of member classifiers corresponds to a cost in terms of resources, especially in DL. The paper is clear and readable. There is no technical novelty.

---

> ### Author Response · Authors · 2022-11-18
> **Response to Reviewer zrP4**
>
> Dear reviewer, thanks for your comments. We find that you misunderstood the evaluation metric in our experiments. We will explain this shortly. Before we answer each concern, we list the contents of this response as follows:
>
> 1. clarification on the evaluation metric AURC;
> 2. clarification on the sufficient number of member models;
> 3. clarification on the choice of the backbone model;
> 4. defense of this paper's novelty.
>
> Following are our answers to each concern.
>
> > Table 1: it is not clear whether the results are presented with the same level of coverage. If the coverage level is different, it is difficult to compare the values. (It is recommended to add the coverage in the table, or to specify and use a fixed coverage).
>
> The coverage level is not needed in Table 1. In Table 1, the evaluation metric AURC (Area Under Risk-coverage Curve) is not specific to certain coverage. Mathematically, for any selective classifier $(f, g)$, since we can plot the risk-coverage curve, the selective risk can be regarded as a function of coverage $R(f,g; \phi)$. Thus, AURC of $(f, g)$ is $\int_0^1 R(f,g; \phi) \mathrm{d}\phi$, which can be interpreted as the average selective risk overall coverage. It is obvious that AURC is not specific to certain coverage; hence, the coverage level is not needed in Table 1.
> We use AURC rather than selective risks under a range of coverage as the evaluation metric because
>
> 1. AURC is comprehensive and concise;
> 2. The selective risks under a range of coverage is not concise since we need to show them in a list;
> 3. we need a concise evaluation metric, considering the large amount of experiments.
>
> > What is a sufficient number of ensemble members?
>
> Two to five ensemble members are sufficient since they provide good performance (see Figure 6) and do not cost too much computation resources.
>
> > What is the accuracy of the model and coverage at M=20 for SR?
>
> The selective risks of the model at M=20 for SR under a series of coverage are listed below, where *ind.* stands for the individual model. The dataset is CIFAR-10. The means and standard deviations are estimated over 3 trials.
>
> | coverage | selective risk (M=20) | selective risk (ind.) |
> | -------- | --------------------- | --------------------- |
> | 1.00     | 5.16$\pm$0.08         | 5.97$\pm$0.10         |
> | 0.90     | 1.49$\pm$0.05         | 2.13$\pm$0.04         |
> | 0.80     | 0.38$\pm$0.02         | 0.66$\pm$0.05         |
> | 0.70     | 0.14$\pm$0.02         | 0.28$\pm$0.03         |
> | 0.60     | 0.09$\pm$0.01         | 0.17$\pm$0.05         |
> | 0.50     | 0.10$\pm$0.00         | 0.15$\pm$0.04         |
> | 0.40     | 0.10$\pm$0.00         | 0.17$\pm$0.06         |
> | 0.30     | 0.10$\pm$0.00         | 0.18$\pm$0.11         |
> | 0.20     | 0.12$\pm$0.02         | 0.17$\pm$0.13         |
> | 0.10     | 0.07$\pm$0.05         | 0.13$\pm$0.12         |
>
> > The big backbone is designed to have twice as many filters in every convolutional layer (VGG-16) why the authors didn't consider a ResNet-50/101?
>
> VGG-16 is used as the standard backbone model in multiple previous works [3][4][5][6], and we follow the choice of previous works to make fair comparisons.
>
> > there is no technical novelty, the contributions are only marginally significant.
>
> The main contribution of this work is not the method but the in-depth understanding (the proof and the assumptions) of why the ensemble works in selective classification, which is not incremental. Please refer to our reply to the last question of Reviewer fKnJ, which elaborates on the challenges in the proof.
>
> [3] Yonatan Geifman and Ran El-Yaniv. Selective classification for deep neural networks. Advances in Neural Information Processing Systems, 2017.
>
> [4] Yonatan Geifman and Ran El-Yaniv. SelectiveNet: A Deep Neural Network with an Integrated Reject Option. In ICML, 2019
>
> [5] Ziyin Liu, Zhikang Wang, Paul Pu Liang, Russ R Salakhutdinov, Louis-Philippe Morency, and Masahito Ueda. Deep gamblers: Learning to abstain with portfolio theory. Advances in Neural Information Processing Systems, 2019.
>
> [6] Lang Huang, Chao Zhang, and Hongyang Zhang. Self-adaptive training: beyond empirical risk minimization. Advances in Neural Information Processing Systems, 2020.

---

### Official Review · Reviewer_X5Ed · 2022-10-24

**Confidence:** 4
**Correctness:** 3
**Technical Novelty And Significance:** 2
**Empirical Novelty And Significance:** 3
**Recommendation:** 5

**Clarity, Quality, Novelty And Reproducibility:**

Overall, the paper is reasonably clear, and the results are novel to the best of my knowledge.

As mentioned above, I think the major issue is with the scope of the results provided in this paper. Although the results are relevant, the limited application of the theory (to highly specific types of ensembles) and the lack of follow-up on the limitations of the results (ensembling improves only up to a threshold $\phi_0$; assumptions can be violated; etc.) limit the impact of this work.

**Strength And Weaknesses:**

# Strengths
- The paper is well structured and motivated; the literature review and general positioning of this paper is clear and well situated within the greater context of the field.
- The paper addresses an important task with clear applications to real life problems (can we automatically detect when to abstain from predicting), and analyzes a well-known, simple technique (ensembling) within this new context.
- The authors conclusively show under intuitive hypotheses that ensembling will improve selective prediction.
- Empirical results confirm that the hypotheses are most of the time reasonable; furthermore, empirical results show that ensembling for selective prediction is beneficial even beyond what theoretical results are provided.

# Weaknesses
- I believe the greatest issue with this work is that the theoretical and empirical results could be more detailed, as the current statements would benefit from greater investigation. For example:
  * (Thm. 2) How does $\phi_0$ depend on various ensemble characteristics? Can we bound it away from zero? Under which conditions?
  * Similarly, I think a deeper analysis of what causes the MRPC dataset to violate the assumptions (which datapoints, and understanding why) would significantly strengthen the paper, especially as the scope of the theoretical results is somewhat limited (next point).
- The scope of the theoretical results is limited to deep ensembles (i.e., ensembles whose predicted probability vector is obtained via a softmax). It wasn't very clear to me whether this assumption is required on top of the three assumptions detailed earlier -- if so, could the authors clarify why this is?
- I found some of the notation unclear -- in some cases, the quantities being manipulated are vectors (of length = number of classes), but numerical estimates are scalars. For example, the standard deviation in $\S$ 6.1. Similarly, is $R_\text{ind}$ in Thm. 2 the average selective risk of all ensemble members?

Minor:
- I would recommend reordering Figure 2, so that each subfigure compares different models on a single dataset.
- This paper could benefit from some quick SPAG proofreading, as there are several minor grammar mistakes throughout the paper.

**Summary Of The Paper:**

The authors analyze ensembles in the context of selective prediction. The authors show that under loose assumptions, a selective classifier using the probability assigned to the predicted class to decide whether to abstain from predicting can be improved by ensembling $n$ copies of it (differing only in their random seed).

The three assumptions required are, paraphrased:
- A1: When all models agree on a prediction, they are less likely to be incorrect than when they disagree.
- A2: Each individual model has a non-zero probability of assigning predicting a class with probability one (implying that models can be overconfidently incorrect).
- A3: The joint PDF of all ensemble members is bounded on ambiguous queries (no prediction has PDF = $+\infty$).

These assumptions in hand, the authors show (Prop 1.) that on ambiguous queries, an ensemble _cannot_ assign probability 1 to any class. It follows that (Thm. 2) ensembling will improve the selective risk up to a coverage $\phi_0 > 0$.

The authors then evaluate their assumptions on vision and NLP datasets, as well as different ensembling techniques for selective prediction (including, but not restricted to, deep ensembles).

**Summary Of The Review:**

This is an interesting paper addressing an important question in the ML community. However, this paper could be significantly improved by probing the derived theoretical and empirical results in much more detail. As it stands, the import of this work remains unclear, and it is difficult to place the specific results and limitations within the greater context of the selective prediction problem.

That being said, I believe that these weaknesses may be easily resolved, making this paper a valuable contribution to the field.

---

> ### Author Response · Authors · 2022-11-18
> **Response to Reviewer X5Ed (3/3)**
>
> > I would recommend reordering Figure 2, so that each subfigure compares different models on a single dataset.
>
> This is a good suggestion. Although comparing different models is not the original goal of this figure, we would like to show these results. However, because this reordered arrangement of subfigures makes some of the curves twisted together and crowded, we will present the results by **tables** in Appendix D of the revision (Table 3 on Page 20 and Table 4 on Page 21). A demo of these tables is as follows, which shows the selective risks of ensembles under coverage 10%-100% on CIFAR-100.
>
> In addition, since the original Figure 2 provides a clear comparison between the ensemble and the individual model in multiple settings, which is necessary for our conclusion, this figure will be preserved in the revision.
>
> Table: the selective risks of ensembles under coverage 10%-100% on CIFAR-100. The means and standard deviations are calculated over three trials. The best entries and those that overlap with the best entries are marked in bold.
> |  coverage | SR ensemble | Gambler ensemble | SAT ensemble | Reg-curr ensemble |
> | - | - | - | - | - |
> | 100| **24.66$\pm$0.08** | 25.50$\pm$0.05 | 25.23$\pm$0.13 | 25.70$\pm$0.09 |
> | 90 | **19.15$\pm$0.15** | 19.88$\pm$0.05 | 19.77$\pm$0.28 | 20.16$\pm$0.14 |
> | 80 | **14.32$\pm$0.22** | 15.75$\pm$0.09 | 15.00$\pm$0.20 | 15.22$\pm$0.07 |
> | 70 | **9.78$\pm$0.13** | 12.11$\pm$0.18 | 10.29$\pm$0.24 | 10.41$\pm$0.38 |
> | 60 | **5.81$\pm$0.06** | 8.89$\pm$0.16 | 6.43$\pm$0.20 | 6.58$\pm$0.27 |
> | 50 | **2.95$\pm$0.04** | 6.22$\pm$0.10 | 3.41$\pm$0.15 | 3.45$\pm$0.05 |
> | 40 | **1.40$\pm$0.13** | 4.37$\pm$0.06 | 1.96$\pm$0.13 | 1.74$\pm$0.11 |
> | 30 | **0.75$\pm$0.05** | 2.67$\pm$0.01 | 1.13$\pm$0.02 | 0.89$\pm$0.06 |
> | 20 | **0.62$\pm$0.06** | 1.91$\pm$0.04 | **0.72$\pm$0.06** | **0.62$\pm$0.04** |
> | 10 | 0.33$\pm$0.09 | 1.42$\pm$0.16 | 0.57$\pm$0.09 | **0.13$\pm$0.05** |
>
> > This paper could benefit from some quick SPAG proofreading, as there are several minor grammar mistakes throughout the paper.
>
> This is a good suggestion, and we will correct grammar mistakes in the revision.

---

> > ### Comment · Reviewer_X5Ed · 2022-12-06
> > **Rebuttal acknowledgement**
> >
> > Thank you for your detailed response! This additional discussion significantly strengthens the submission.
> >
> > However, as the main contribution of this work are the theoretical results, I still think reorganizing the paper so as to provide a more in-depth discussion of the high-level details of the proof is necessary. I appreciate the clarification on the intuition provided in the top comment, but would also like to see a discussion of why the specific form of Assumption 3 comes into play, for example; the additional material regarding $\phi_0$ would, in my opinion, also deserve to move to the main text.

---

> > > ### Author Response · Authors · 2022-12-08
> > > **Response to Reviewer X5Ed's reply**
> > >
> > > Dear reviewer, thanks for your constructive comments. We agree with your advice. In the future revision, we will add materials regarding $\phi_0$'s bound to our paper and reorganize our paper to provide a more in-depth discussion of the high-level details of the proof.

---

> ### Author Response · Authors · 2022-11-18
> **Response to Reviewer X5Ed (2/3)**
>
> > I think a deeper analysis of what causes the MRPC dataset to violate the assumptions (which datapoints, and understanding why) would significantly strengthen the paper
>
> We found that the violation of the assumptions on MRPC dataset is due to the insufficient training epochs of the classifiers. In our experiments, following [1], we set the training epoch of the classifiers to be three for text classification. However, three epochs turn out to be insufficient in our follow-up experiments. We retrain the classifiers with more epochs and find the classifiers undergo underfitting in the first three epochs. At the same time, the assumptions are violated. On the contrary, with 10 training epochs, the classifiers fit the training set well, the assumptions are satisfied, and the performances of ensemble models are improved. Thus, it is the insufficient training epochs of the classifiers that causes the MRPC dataset to violate the assumptions. We will retrain classifiers with 10 epochs on MRPC and update the experimental results in the revision (Figure 1, Figure 2, Table 1, Figure 5).
>
> This incident indicates that our assumptions catch the underlying cause of the superiority of the ensemble.
>
> > The scope of the theoretical results is limited to deep ensembles (i.e., ensembles whose predicted probability vector is obtained via a softmax). It wasn't very clear to me whether this assumption is required on top of the three assumptions detailed earlier -- if so, could the authors clarify why this is?
>
> We are unsure if you wonder whether our analysis depends on the softmax. If so, the answer is no, and our analysis requires only that each member model produces a predictive probability distribution.
>
> If you wonder whether we use the deep ensemble that combines the members' predictive probability distributions by averaging rather than other kinds of ensembles, such as voting, the answer is yes. This choice is because of the good performance of the deep ensemble in practice (see [2] and the experimental results in this paper). We have tried other ensembles, such as the ensemble that combine members' logits by averaging; those ensembles work too, but not as well as the deep ensemble.
>
> > in some cases, the quantities being manipulated are vectors (of length = number of classes), but numerical estimates are scalars. For example, the standard deviation in §§ 6.1.
>
> Vectors and scalars are not mixed in this paper. The $(\cdot)^2$ in the definition of STD (in Section 6.1) is $\Vert \cdot\Vert^2$, so the STD is a scalar.
>
> > is $R_{\mathrm{ind}}$ in Thm. 2 the average selective risk of all ensemble members?
>
> No, $R_{\mathrm{ind}}$ is the selective risk of any given individual model. This is unclear in the paper and will be clarified in the revision (Theorem 2, Page 6).
>
> [1] Ji Xin, Raphael Tang, Yaoliang Yu, and Jimmy Lin. The art of abstention: Selective prediction and error regularization for natural language processing. In Proceedings of the 59th Annual Meeting of the Association for Computational Linguistics and the 11th International Joint Conference on Natural Language Processing, 2021.
>
> [2] Balaji Lakshminarayanan, Alexander Pritzel, and Charles Blundell. Simple and Scalable Predictive Uncertainty Estimation using Deep Ensembles. In NeurIPS, 2017.

---

> ### Author Response · Authors · 2022-11-18
> **Response to Reviewer X5Ed (1/3)**
>
> Dear reviewer, thanks for your valuable and constructive comments. The contents of this response are as follows:
>
> 1. the result of follow-up study, including bounding $\phi$ and analyzing the violation of assumptions on MRPC;
> 2. clarification on the theoretical analysis' scope;
> 3. clarification on the concerns about notations;
> 4. response to the advice on Figure 2;
> 5. response to grammar mistakes in this paper.
>
> Following are our answers to each concern.
>
> > (Thm. 2) How does $\phi_0$ depend on various ensemble characteristics? Can we bound it away from zero? Under which conditions?
>
> These problems are valuable and worth further study.
>
> We can bound $\phi_0$ away from zero. We extend our analysis and provide a lower bound of $\phi_0$ in Appendix G of the revision. However, we do not obtain a closed-form expression of the lower bound but express it as an optimization problem's solution. Thus, we provide a search algorithm to calculate $\phi_0$'s value. Below is a summary of our results.
>
> **Preliminaries:** Let $R(\phi)$ be the selective risks of the individual model under coverage $\phi$; $\phi(\tau)$ is the coverage of the individual model given the confidence threshold $\tau$; $C$ is the confidence of the individual model; $A$/$D$ is the event that the input sample is ambiguous/definite. Assume that
> 1. we have all knowledge about the individual model, e.g., the mapping from $\tau$ to $\phi$ is known;
> 2. $R$ is a monotone increasing function of coverage $\phi$, and $\phi$ is a monotone decreasing function of the confidence threshold $\tau$;
> 3. the following are given: an oracle that tells whether a sample is definite; the number of classes $K$; the number of member models $M$;
> and $B$, the upper bound of $p_{\Pi_1^k, ..., \Pi_M^k}(\cdot|A)$ for all $k\in\{ 1, 2, ..., K\}$.
>
> **The Lower Bound of $\phi_0$:** We can obtain the lower bound of $\phi_0$ by solving the following optimization problem.
>
> $ \max_{\tau,\tau^*} \ \phi(\tau) $,
>
> subject to $ \phi(\tau) = P(C\ge\tau^*, D)$,
>
> and $P(A)\cdot \beta(1-\tau^*)^M + P(Err, D, C\ge\tau^*) < P(Err, C\ge\tau)$;
>
> or equivalently, solving the following optimization first and then calculating the corresponding coverage of the optimal solution,
>
> $ \min_{\tau,\tau^*} \ \tau $,
>
> subject to $ \phi(\tau) = P(C\ge\tau^*, D)$,
>
> and $P(A)\cdot \beta(1-\tau^*)^M + P(Err, D, C\ge\tau^*) < P(Err, C\ge\tau)$;
>
> **Algorithm:** Since $\tau^*$ is determined by $\tau$ (the first constraint of the optimization problem above), this optimization problem can be reduced to a one-dimensional search problem. We adopt a binary search to solve this problem. The revision provides details of the algorithm in Appendix G (Page 26).
>
> **An Example:** To show that our algorithm works in reality, we run this algorithm on CIFAR-10, using the same individual model as this paper's experiments. In this example,$K = 10$, $M = 5$, the oracle is implemented by another ensemble with two individual models (the oracle outputs *True* if and only if the STD over member models' predictive distributions < $10^{−3}$). We do not estimate $B$ but try several hypothetical values of $B$ to see at what $B$ the lower bound of $\phi_0$ is big. With different $B$s, we obtain different lower bounds of $\phi_0$ as the following table shows. We can see that when $B\le 10^8$, the lower bound of $\phi_0$ is greater than 50%, which indicates that our algorithm may be robust to the choice of $B$.
>
> | B | 1 | 10 | $10^2$ | $10^3$ | $10^4$ | $10^5$ | $10^6$ | $10^7$ | $10^8$ | $10^9$ | $10^{10}$ | $10^{11}$ | $10^{12}$ | $10^{13}$ | $10^{14}$ |
> | - |    -    |     -   | -       |  -      |    -    |    -    |    -    |    -    |       - |    -    |    -    |    -    |    -    |    -    |    -    |
> | lower bound of $\phi_0$ | 0.737 | 0.737 | 0.737 | 0.737 | 0.736 | 0.735 | 0.729 | 0.705 | 0.599 | 0.00 | 0.00 | 0.00 | 0.00 | 0.00 | 0.00 |
>
> This example also indicates the relationship between the ensemble's diversity and its selective classification performance. Since an ensemble with a smaller $B$ seems to have more diversity over ambiguous samples, the result in the table suggests that as long as the ensemble has enough diversity over ambiguous samples, the ensemble is guaranteed to have a lower selective risk than the individual model under a considerable range of coverage.

---

### Official Review · Reviewer_fKnJ · 2022-10-27

**Confidence:** 3
**Correctness:** 4
**Technical Novelty And Significance:** 2
**Empirical Novelty And Significance:** 2
**Recommendation:** 5

**Clarity, Quality, Novelty And Reproducibility:**

The proof in the paper is hard to read currently. It would also be good to provide some intuition of why we should expect the proof to go through. The ideas are somewhat incremental and I don't think the proofs provide any new technical challenges.

**Strength And Weaknesses:**

The idea of combining ensembles and selective classification is interesting.

In assumption 2, why do we need the definite samples probability at confidence approaching 1 to be non-zero? It seems that the entire advantage of ensembles is coming out of the ambiguous samples.

For the ambiguous samples, assumption 2 says that there are a few ambiguous samples which are predicted with confidence by individual models. Why it can’t be the case that all the models predict the wrong label with confidence approaching 1?

It is very hard to understand assumption 3 in the paper and it is not clear what the intention of that assumption is. It would be good to explain that in the paper. Also, the authors have verified assumptions 1 and 2 but have not commented on the applicability of assumption 3 for practical settings. Can the authors please comment on that?

**Summary Of The Paper:**

Selective classifiers have the ability to abstain on certain datapoints and consist of two functions - the classification function and the selection function. This paper considers ensembes of selective classifiers by averaging the probability distributions of the classifiers and also defining the selection function in a certain way. They provide some assumptions and theoretically prove that under those assumptions, the ensemble of selective classifiers has better selective risk than each of the individual models for certain values of coverage. They also verify these assumptions on a few datasets. They also find that the simplest ensemble of classifiers with maximum probability as the selection function has the best performance compared to the previous selective classification approaches.

**Summary Of The Review:**

I think the idea is interesting but the paper lacks clarity and is not well written.

---

> ### Author Response · Authors · 2022-11-18
> **Response to Reviewer fKnJ (2/2)**
>
> > It is very hard to understand assumption 3 in the paper and it is not clear what the intention of that assumption is. It would be good to explain that in the paper. Also, the authors have verified assumptions 1 and 2 but have not commented on the applicability of assumption 3 for practical settings. Can the authors please comment on that?
>
> Assumption 3 is related to the definition of ambiguous samples. Consider an ensemble with two members $\theta_1$ and $\theta_2$. We denote their predictive probability for class $k$ as $\Pi_1^k$ and $\Pi_2^k$ $(k\in${$1,2,...,K$}), respectively. On definite samples, both $\theta_1$ and $\theta_2$ provide the same predictive probability distribution ($\Pi_1^k = \Pi_2^k, \forall k\in${$1,2,...,K$}). Thus, for all $k$, $p_{\Pi_1^k, \Pi_2^k}(\cdot|D)$, the joint distribution of $\Pi_1^k$ and $\Pi_2^k$ given the input samples being definite, collapses to {$(\lambda, \lambda) | \lambda\in[0, 1]$}. In other words,
> $p_{\Pi_1^k, \Pi_2^k}(u,v|D) = +\infty$ if $(u, v)\in${$(\lambda, \lambda) | \lambda\in[0, 1]$}; otherwise, $p_{\Pi_1^k, \Pi_2^k}(u,v|D)=0$. On the contrary, ambiguous samples do not have such a property. We intensify this by Assumption 3 to provide a good analytical property of ambiguous samples.
>
> Furthermore, Assumption 3 reflects the diversity of the ensemble over ambiguous samples. Still consider the example above. If the predictions of $\theta_1$ and $\theta_2$ are sure to coincide, i.e., the ensemble model has no diversity, then the PDF of $\Pi_1^k$ and $\Pi_2^k$ is unbounded. Conversely, if the PDF of $\Pi_1^k$ and $\Pi_2^k$ is bounded, then the predictions of the member models are diverse, and the ensemble has diversity. Thus, Assumption 3 depicts the diversity of the ensemble over ambiguous samples.
>
> It is well known that the randomization-based ensemble has diversity. Since the ensemble does not have diversity over definite samples, it must have diversity over ambiguous samples. Thus, we do not provide experimental verification for Assumption 3.
>
> > The proof in the paper is hard to read currently. It would also be good to provide some intuition of why we should expect the proof to go through.
>
> The proof is somewhat complex, but its intuition is straightforward. In a word, the intuition is that because the ensemble avoids being overconfident over ambiguous samples, the ensemble has a lower selective risk. Details of the intuition are as follows:
>
> 1. the individual model is **not** "modest" over both ambiguous samples and definite samples (Assumption 2);
> 2. by contrast, based on Assumption 3, we prove that the ensemble provides modest confidence to ambiguous samples (Proposition 1). In addition, the confidence of definite samples remains the same throughout ensembling (due to the definition of definite samples);
> 3. thus, when confidence approaches 1, as long as the classifier's error rate over definite samples is lower than the error rate over ambiguous samples (Assumption 1), the individual model suffers more error predictions that come with the ambiguous samples than the ensemble. Based on this, we prove that the selective risk drops under a range of coverage via ensembling (Theorem 2).
>
> This intuition is added to Appendix B in the revision (Pages 11-12).
>
> > The ideas are somewhat incremental and I don't think the the proofs provide any new technical challenges.
>
> We would like to emphasize that the main contribution of this work is not the method but the understanding (the proof and the assumptions) of why the ensemble works in selective classification. Below is the defense of our contribution.
>
> First, the proof is not incremental. The proof is challenging and totally different from the previous proofs on the ensemble in classification or regression. The proofs of the ensemble in standard classification are based on assumptions of unbiased, uncorrelated, and identically distributed estimation errors for the posterior probability distribution, which are not adopted in this paper. As for the proof in regression, the proof makes use of the convexity of the square loss. However, the metrics of selective classification, e.g., the selective risk, are non-convex, so the proof in regression cannot be adapted to selective classification.
>
> Second, we propose some reasonable assumptions which support our proof process and are experimentally verified. These assumptions are critical factors for the success of the ensemble model. Alone with the proof, the assumptions provide an in-depth understanding of why the ensemble works in selective classification, that is, **because the ensemble avoids being overconfident over ambiguous samples, the ensemble has a lower selective risk**.

---

> ### Author Response · Authors · 2022-11-19
> **Response to Reviewer fKnJ (1/2)**
>
> Dear reviewer, thanks for your careful and valuable comments. The contents of this response are as follows:
>
> 1. clarification on Assumption 2;
> 2. clarification on Assumption 3;
> 3. the proof's intuition;
> 4. defense of this paper's novelty.
>
> Following are our answers to each concern.
>
> > In assumption 2, why do we need the definite samples probability at confidence approaching 1 to be non-zero? It seems that the entire advantage of ensembles is coming out of the ambiguous samples.
>
> Actually, the advantage of ensembles comes from both the ambiguous and definite samples, and we cannot prove Theorem 2 without any assumption that depicts the confidence of the definite samples. Consider a counterexample:
>
> 1. the dataset has 50% definite samples and 50% ambiguous samples;
> 2. an individual model $(f,g)$ gives confidence 0.5-0.6 to the definite samples and gives confidence 0.6-1.0 to the ambiguous samples;
> 3. the classifier $f$'s error rate is 0.001 over any definite sample and 0.1 over any ambiguous sample.
>
> In this case, ambiguous samples' confidence is always higher than definite samples' confidence. Therefore, when the coverage is less than 50%, $(f, g)$ rejects all definite samples but makes predictions on ambiguous samples. Hence, the selective risk is 0.1 under any coverage below 50%. Now consider the ensemble. It is easy to see that the range of confidence provided by the ensemble for ambiguous samples is a subset of that provided by the individual, i.e. [0.6, 1.0], whilst confidence of the definite samples remains the same throughout ensembling (due to the definition of definite samples). Thus, ambiguous samples' confidence is still always higher than definite samples' confidence, and the selective risk remains 0.1 under any coverage below 50%. This example suggests that the performance of the selective classifier depends on the relative ranking of all samples' confidence instead of part of the samples' confidence. Thus, we need an assumption that depicts both definite and ambiguous samples' confidence.
>
> This paper's assumption is a little strong for the convenience of proof. Although the experimental results suggest that this assumption is the actual behavior of SR models, we guess the assumption can be relaxed while the conclusion keeps the same. This relaxation is an interesting direction for future work.
>
> > For the ambiguous samples, assumption 2 says that there are a few ambiguous samples which are predicted with confidence by individual models. Why it can’t be the case that all the models predict the wrong label with confidence approaching 1?
>
> The case is possible and does not conflict with Assumption 2. Individual models can make the same but wrong prediction on a definite sample with confidence approaching 1, as far as the corresponding error rate satisfies Assumption 1, that is, the classifier's error rate over definite samples is lower than that over ambiguous samples with confidence approaching 1.

---

### Author Response · Authors · 2022-11-18
**General Response: FAQs**

Dear reviewers, thanks for your valuable comments. In this general response, we will answer some frequently asked questions (FAQs), including

1. it is difficult to understand Assumption 3 (or its description);
2. provide some intuition of why we should expect the proof to go through
3. the idea is incremental, no novelty or challenge.

Below are our answers to each question.

> 1. it is difficult to understand Assumption 3 (or its description)

Assumption 3 is related to the definition of ambiguous samples. Consider an ensemble with two members $\theta_1$ and $\theta_2$. We denote their predictive probability for class $k$ as $\Pi_1^k$ and $\Pi_2^k$ $(k\in${$1,2,...,K$}), respectively. On definite samples, both $\theta_1$ and $\theta_2$ provide the same predictive probability distribution ($\Pi_1^k = \Pi_2^k, \forall k\in${$1,2,...,K$}). Thus, for all $k$, $p_{\Pi_1^k, \Pi_2^k}(\cdot|D)$, the joint distribution of $\Pi_1^k$ and $\Pi_2^k$ given the input samples being definite, collapses to {$(\lambda, \lambda) | \lambda\in[0, 1]$}. In other words, $p_{\Pi_1^k, \Pi_2^k}(u,v|D) = +\infty$ if $u=v\in[0,1]$, i.e., $(u, v)\in$ {$(\lambda, \lambda) | \lambda\in[0, 1]$}; otherwise, $p_{\Pi_1^k, \Pi_2^k}(u,v|D) = 0$. On the contrary, ambiguous samples do not have such a property. We intensify this by Assumption 3 to provide a good analytical property of ambiguous samples.

Furthermore, Assumption 3 reflects the diversity of the ensemble over ambiguous samples. Still consider the example above. If the predictions of $\theta_1$ and $\theta_2$ are sure to coincide, i.e., the ensemble model has no diversity, then the PDF of $\Pi_1^k$ and $\Pi_2^k$ is unbounded. Conversely, if the PDF of $\Pi_1^k$ and $\Pi_2^k$ is bounded, then the predictions of the member models are diverse, and the ensemble has diversity. Thus, Assumption 3 depicts the diversity of the ensemble over ambiguous samples.

It is well known that the randomization-based ensemble has diversity. Since the ensemble does not have diversity over definite samples, it must have diversity over ambiguous samples. Thus, we do not provide experimental verification for Assumption 3.

> 2. provide some intuition of why we should expect the proof to go through

The proof's intuition is straightforward. In a word, because the ensemble avoids being overconfident over ambiguous samples, it achieves a lower selective risk. More specifically:

1. the individual model is **not** "modest" over both ambiguous samples and definite samples (Assumption 2);
2. by contrast, based on Assumption 3, we prove that the ensemble provides modest confidence to ambiguous samples (Proposition 1). In addition, the confidence of definite samples remains the same throughout ensembling (due to the definition of definite samples);
3. thus, when confidence approaches 1, as long as the classifier's error rate over definite samples is lower than the error rate over ambiguous samples (Assumption 1), the individual model suffers more error predictions that come with the ambiguous samples than the ensemble. Based on this, we prove that the selective risk drops under a range of coverage via ensembling (Theorem 2).

This intuition is added to Appendix B in the revision (Pages 11-12).

> 3. the idea is incremental, no novelty or challenge.

We would like to emphasize that the main contribution of this work is the understanding (the proof and the assumptions) of why the ensemble works in selective classification. Below is the defense of our contribution.

First, the proof is not incremental. The proof is challenging and totally different from the previous proofs on the ensemble in classification or regression. The proofs of the ensemble in standard classification are based on assumptions of unbiased, uncorrelated, and identically distributed estimation errors for the posterior probability distribution, which are not adopted in this paper. As for the proof in regression, the proof makes use of the convexity of the square loss. However, the metrics of selective classification, e.g., the selective risk, are non-convex, so the proof in regression cannot be adapted to selective classification.

Second, we propose some reasonable assumptions which support our proof process and are experimentally verified. These assumptions are critical factors for the success of the ensemble model. Alone with the proof, the assumptions provide an in-depth understanding of why the ensemble works in selective classification, that is, **because the ensemble avoids being overconfident over ambiguous samples, the ensemble has a lower selective risk**.

---

### Author Response · Authors · 2022-11-18
**General Response: Major Changes in the Revision**

Dear reviewers, thanks for your valuable comments. In this general response, we will address major changes in the revision, which are summarized as follows:

1. We update the experimental results on MRPC dataset (Figures 1, 2, 5, and Table 1). In the first submission, the result on MRPC violates the assumptions. We find this is because the training process does not converge, as the MPRC dataset is too small. By only increasing training epochs, the result is now in line with our assumptions, and the performance is improved. This incident indicates that our assumptions catch the underlying cause of the superiority of the ensemble.
2. We extend our analysis and provide a lower bound of $\phi_0$ (Theorem 2) in Appendix G (Page 24).
3. The description of Assumption 3 (Page 5) is updated to make Assumption 3 more clear.
4. The proof's intuition is presented before our theoretical results and proofs (Section 5.2 and Appendix B).

---

### Decision · Program_Chairs · 2023-01-20

**Decision:**

Reject

**Justification For Why Not Higher Score:**

* Current structure of the paper fails to effectively deliver the main contributions (theoretical analysis/proof techniques)
* Clarity/notation issues around key components of the paper (e.g., assumptions)

**Justification For Why Not Lower Score:**

N/A

**Metareview: Summary, Strengths And Weaknesses:**

The reviewers and meta reviewer all carefully checked and discussed the rebuttal. They thank the authors for their response and their efforts during the rebuttal phase.
The response helped resolve some concerns (e.g., improvement of the experiments with new results for MRPC, together with a few clarifications about notations/assumptions).

The reviewers and meta reviewer all acknowledge that the submission investigates a relevant and important problem (“important task with clear applications to real life problems”, “selective classification is an interesting and important problem”).

At this stage, however, there are still some arguably important aspects that would benefit from further consolidations, in particular:

* As acknowledged by the authors, the core contribution of the paper lies in the theoretical analysis and the new proof technique(s). Right now, this is not properly surfaced by the manuscript. A substantial reorganization of the manuscript around that material would help deliver the main messages.

* In parallel to the point above, several reviewers raised concerns about the clarity of the notation, assumptions and other technical elements around the theoretical results (the changes made during the rebuttal phase go in the right direction). Since those represent the core results of the paper, they need to be more clearly exposed.

Because of the extremely competitive landscape of the submissions this year, the paper remains under the cut and has ultimately not been selected for acceptance.
We are convinced that the suggestions above will help strengthen the paper for a future resubmission, which the reviewers and meta reviewer all encourage.